# Janus decellularized membrane with anisotropic cell guidance and anti-adhesion silk-based coatings for spinal dural repair

Xuewei Bi[1,2,3,4,6], Zhinan Mao[4,5,6], Linhao Li [1,2] ✉, Yilin Zhang[2], Lingbing Yang[2], Sen Hou[2], Juan Guan [5], Yufeng Zheng [4], Xiaoming Li [1,2,3] ✉ & Yubo Fan [1,2,3] ✉

The repair of soft tissues with anisotropic structures, such as spinal dura mater, requires the use of biomaterials to guide tissue directional growth while minimizing epidural fibrotic adhesion. Herein, we construct the Janus small intestinal submucosa (SIS) via silk-based hydrogel coatings, which provides extracellular matrix-mimicking features and anti-adhesion performance for spinal dural defect repair. We demonstrate that the silk fibroin and methacrylated silk fibroin (SilMA) composite microgroove hydrogel coating at the inner surface via water vapor annealing treatment exhibits excellent structure stability, stable attachment to SIS substrate, and shows orientated cell morphology and extracellular matrix produced by fibroblasts, good histocompatibility and promotes the polarization of macrophages towards the anti-inflammatory phenotype. The methacrylated hyaluronic acid and SilMA composite coating outer surface serves as favorable physical barrier shows effective resistance to protein adsorption, cell and tissue adhesion, and can mitigate fibrosis reactions. Spinal dura mater defect experiments on male rats demonstrate that the Janus SIS simultaneously promotes dural regeneration and inhibits epidural fibrosis.

The spinal dura mater plays a very important role in protecting the spinal cord, preventing cerebrospinal fluid leakage and epidural adhesions[1,2]. In spinal surgery and neurosurgery, spinal dura mater defect leading to cerebrospinal fluid leakage is a common postoperative complication, with an incidence rate of 4%–32% depending on the type of surgery[3,4]. Till today, few materials are developed for spinal dural repair. The commonly used materials alternatives in clinical practice are animal-derived collagen matrix (e.g., DuraGen®), decellularized extracellular matrix (dECM) (e.g., SIS) and synthetic patches (e.g., polyglycolic acid), shortcomings such as limited sources, mismatched degradation rates with tissue regeneration, inflammation and fibrotic adhesions limits their clinical efficacy[5,6]. Importantly, these alternatives do not fully consider the natural extracellular matrix (ECM) characteristics of the spinal dura mater and are unable to deal with the multiple requirements of dural regeneration. The ECM of spinal dura mater is mainly composed of longitudinal and parallel collagen fibers with anisotropy character[7]. To achieve structural regeneration, the material with ECM-mimicking microstructure that

¹Innovation Center for Medical Engineering & Engineering Medicine, Hangzhou International Innovation Institute, Beihang University, Hangzhou, China. ²Key Laboratory of Biomechanics and Mechanobiology of Ministry of Education, Beijing Advanced Innovation Center for Biomedical Engineering, School of Biological Science and Medical Engineering, and with the School of Engineering Medicine, Beihang University, Beijing, China. ³National Medical Innovation Platform for Industry-Education Integration in Advanced Medical Devices (Interdiscipline of Medicine and Engineering), Key Laboratory of Innovation and Transformation of Advanced Medical Devices of Ministry of Industry and Information Technology, Beihang University, Beijing, China. ⁴School of Materials Science and Engineering, Peking University, Beijing, China. ⁵School of Materials Science & Engineering, Beihang University, Beijing, China. ⁶These authors contributed equally: Xuewei Bi, Zhinan Mao. ✉e-mail: linhaoli@buaa.edu.cn; x.m.li@hotmail.com; yubofan@buaa.edu.cn

enable to promote the directional growth of tissue cells is expected[8,9]. Meanwhile, it is necessary to avoid biological accumulation events of proteins and cells on the other side facing away from the tissue, so as to reduce the occurrence of epidural fibrotic adhesion[10,11]. Therefore, it is imperative to develop biomimetic Janus material to promote structural regeneration of the spinal dura mater and prevent epidural fibrosis, which is important for the functional restoration of the spinal dura mater.

SIS, one of the representative dECM materials, has irreplaceable clinical value in the field of soft tissue injury repair due to its inherent bioactivity, non-immunogenicity, and degradability[12]. It could slowly release growth factors such as transforming growth factor-β, vascular endothelial growth factor, and basic fibroblast growth factor during degradation, thereby regulating host cell phenotype and function, ultimately promoting the formation of neo-tissues[13,14]. However, the lack of anisotropic structure and anti-adhesion ability limit their development for spinal dura mater tissue engineering. In this regard, fabricating coatings on the SIS membrane that not only precisely promote the growth of target tissue cells while providing anti-adhesion to the periphery is expected to further improve clinical effectiveness.

Current strategies for the construction of anisotropic structure, including micropatterned substrates and aligned polymeric fibrous matrices are utilized extensively. Due to the precise control of the morphology and size of the ridges/grooves at the microscale, the micropattern substrate allows for regulation of cell behavior[15,16]. Studies have proved that micropattern substrate could induce a well-organized extracellular matrix, promote M2 macrophage polarization and accelerate native-like tissue regeneration[17]. Hydrogels are water-containing three-dimensional networks that are similar to biological soft tissues[18]. Therefore, given the importance of ECM microstructure and the similarity between hydrogels and biotissue, the construction of a hydrogel coating that mimics the structure of biological tissue will be beneficial to humans[19]. However, the prevalent problem of hydrogels that have been developed are susceptible to swelling under physiological conditions, leading to rapid shape deformation[20,21]. Thus, a customized, micropatterned hydrogel coating with an anisotropic structure, biocompatibility, and anti-swelling stability is highly desirable.

Excessive deposition of fibrin, proliferation of the tissue cells (i.e., fibroblasts/myofibroblasts) and deposition of ECM components (i.e., collagen) during the dura mater healing process could lead to epidural fibrosis and subsequent adhesion[22,23]. Various strategies have been studied to prevent epidural adhesions, such as improving the surgery techniques, local or systemic treatment with drugs and using biomaterial-based physical barriers[24,25]. Hyaluronic acid (HA) is a highly hydrated natural macromolecule that exists in human tissues. HA and its derivatives have been used as anti-adhesion barriers in clinical practice for many years[26]. It has been reported that the polyanionic ligand effect of HA has an inhibitory effect on proliferation and migration of fibroblasts and the expression of postoperative fibrosis related cytokines[11,27].

Silk fibroin (SF) plays an increasing role in tissue engineering and regenerative medicine, clinical trials and commercialized medical devices due to its inherent biocompatibility, low immunogenicity, excellent mechanical properties and tunable rates of biodegradability[28,29]. The SF chains are composed of crystalline region interspersed with amorphous region, the crystalline region of which is rich in β-sheets and α-helix structure, and the amorphous region of which is mainly random coils structure[30–32]. The molecular weight (MW) and crystalline content of SF chain, water behavior of the different structures (hydrophilic vs hydrophobic) can strongly influence the SF final properties (degradation, mechanical properties, hydrophily). The control of regeneration process (degumming and dissolving) and post-treatment of materials (methanol/ethanol and water vapor annealing treatment) allows to tune SF's MW and crystallinity[33–35].

In this work, to satisfy the clinical requirements of spinal dura mater regeneration, we design a Janus SIS with an inner photocurable SF-SilMA (SFMA) microgroove hydrogel coating for guiding cell/tissue alignment and an outer methacrylated hyaluronic acid (HAMA)-SilMA hydrogel coating for resisting cell adhesion so as to simultaneously mimic the functions of spinal dura matter in promoting tissue regeneration and preventing epidural fibrosis. Both SFMA microgroove and HAMA-SilMA coatings are fully investigated through physiochemical methodologies, followed by the evaluation of the coatings in vitro effect on cell behavior using NIH3T3 fibroblasts. Meanwhile, the histological assessment of the coatings in vivo is fully conducted. A spinal dura mater defect model on Sprague Dawley (SD) rat is employed to investigate the in vivo performance of Janus SIS on dura mater regeneration and suppression of epidural fibrosis.

## Results

### Design and fabrication of Janus SIS membranes

In this study, the Janus SIS with cell contact guidance and anti-adhesion functions was fabricated by photocuring and micromolding techniques (Fig. 1 and Supplementary Fig. 1). First, the SFMA was coated onto the SIS membrane surface via the layer-by-layer (LbL) self-assembly technique according to our previous study[36,37]. The deposition of double bonds on the SIS surface membrane can be used as a "bridge" to connect subsequent SFMA microgroove and the HAMA-SilMA hydrogel coating (Fig. 1a). Then, a patterned polydimethylsiloxane (PDMS) template with 20 μm width was filled with SFMA solution, and SFMA-SIS membrane was further covered on the above solution. Here, the introduction of SF with higher MW and crystallinity was responsible for structural stability and mechanical resistance for microgroove formation (Supplementary Figs. 2 and 3). After photocured, the inner surface of the SFMA-SIS membrane was decorated with microgroove (spinal dura mater ECM-mimicking features) for guiding cell alignment. The HAMA-SilMA hydrophilic coating was fabricated on the outer surface of the SFMA-SIS membrane to resist protein adsorption and cell adhesion, thereby minimizing fibrotic adhesion. To reduce the swelling of SFMA microgroove hydrogel coating, a 2-min water vapor annealing treatment was adopted to induce the β-sheet formation of SF (Fig. 1b). Finally, the efficacy of Janus SIS membrane on promoting healing and resisting adhesion was verified in a rat spinal dural defect model (Fig. 1c).

### Morphological characterization of Janus SIS membrane

The SilMA, HAMA and SilMA photopolymerization reaction in the presence of lithium phenyl (−2,4,6-trimethylbenzoyl) phosphonate (LAP) photoinitiator is illustrated in Fig. 2a. The surface morphologies of SIS, SFMA- SIS membrane, SFMA microgroove coating before and after water vapor annealing, and HAMA-SilMA coating were evaluated by scanning electron microscopy (SEM), 3D optical scanning, and confocal laser scanning microscopy CLSM (Fig. 2b). The results of SEM and 3D optical scanning images indicated that the surface of SIS has a fibrous and porous structure, and the diameters of the fibers were not uniform. After the deposition of SFMA via LbL self-assembly, the fibers and pores on the surface of SIS disappeared, and the surface became smooth. In the SFMA microgroove coating samples, the surface featured with microgrooves at 20 μm intervals and depths, and there were no changes on the microstructure of SFMA microgroove coating after water vapor annealing. On the other side, the HAMA-SilMA surface was flat. In addition, the cross-section images further demonstrated the asymmetric character of the Janus SIS, and the coatings were concentrated on the surface of the SIS membrane, the interior of the SIS membrane maintained a loose structure without excessive affusion of SF (Fig. 2c). Furthermore, the thickness of the silk-based coatings could be adjusted according to the requirements (Supplementary Fig. 4).

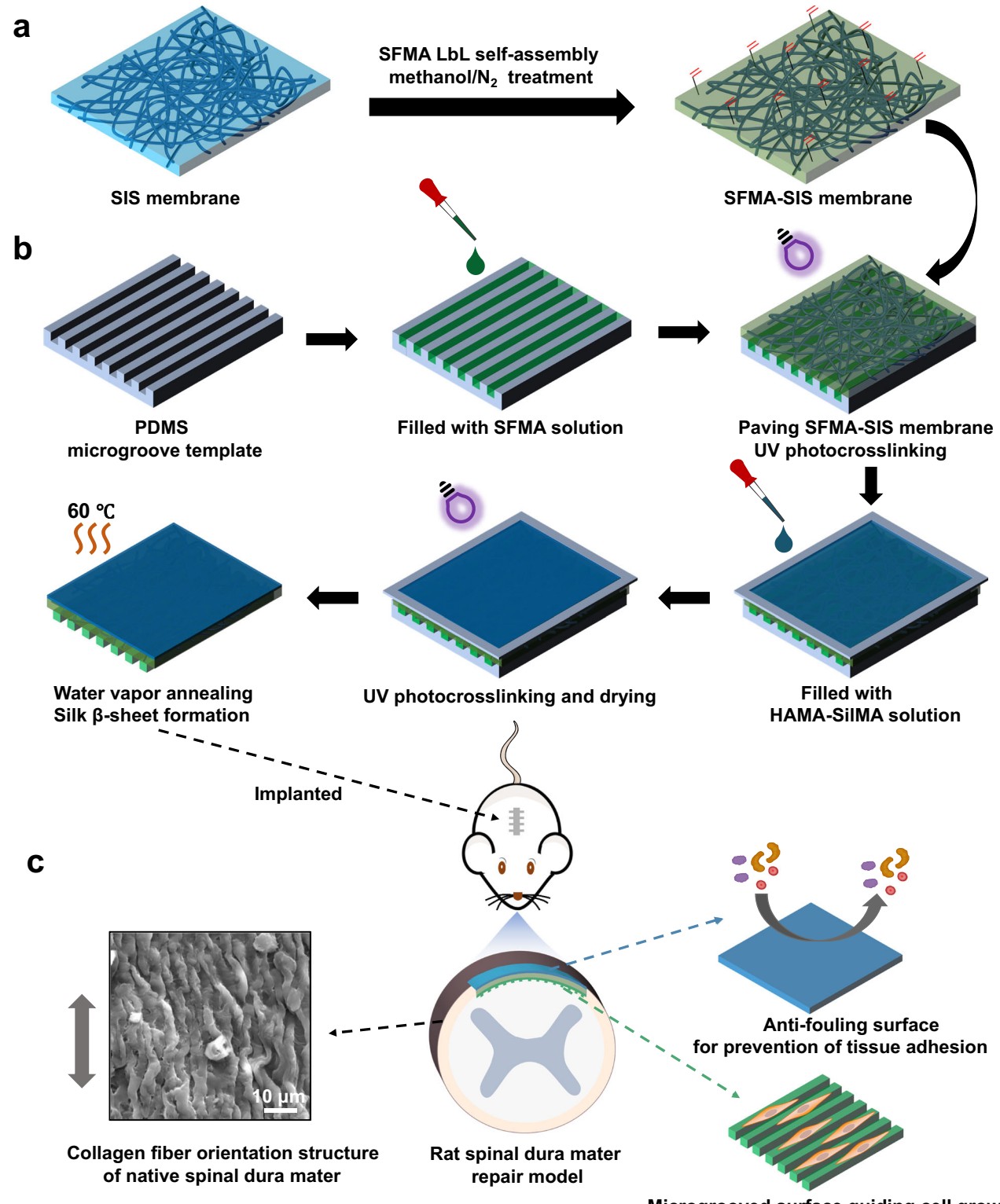

**Fig. 1 | Design and fabrication of Janus SIS membrane for spinal dura mater regeneration. a** Preparation process of the SFMA-SIS: the 1 mg/ml SFMA solution was deposited on the SIS surface via LbL self-assembly; then, the carbon-carbon double bonds on the SIS surface could act as covalent binding sites of functional hydrogel coating. **b** Fabrication process of the SFMA microgroove and HAMA-SilMA hydrogel coatings on SFMA-SIS surface: the PDMS template with 20 μm width microgroove was treated by plasma to endow it with hydrophilic surface, and 60 mg/ml SFMA solution was dripped into the microgroove. Then, SFMA-SIS was covered on the surface of the solution and subjected to ultraviolet light immediately, and the HAMA-SilMA solution was added uniformly on the upper surface of SFMA-SIS and subjected to UV irradiation. After dried, the Janus SIS was stripped off the PDMS mold and treated at 60 °C water vapor for 2 min. **c** In a rat spinal dural defect model, the Janus SIS showed guiding cell growth and preventing tissue adhesion dual functions in vivo, finally, the defect area was repaired in situ.

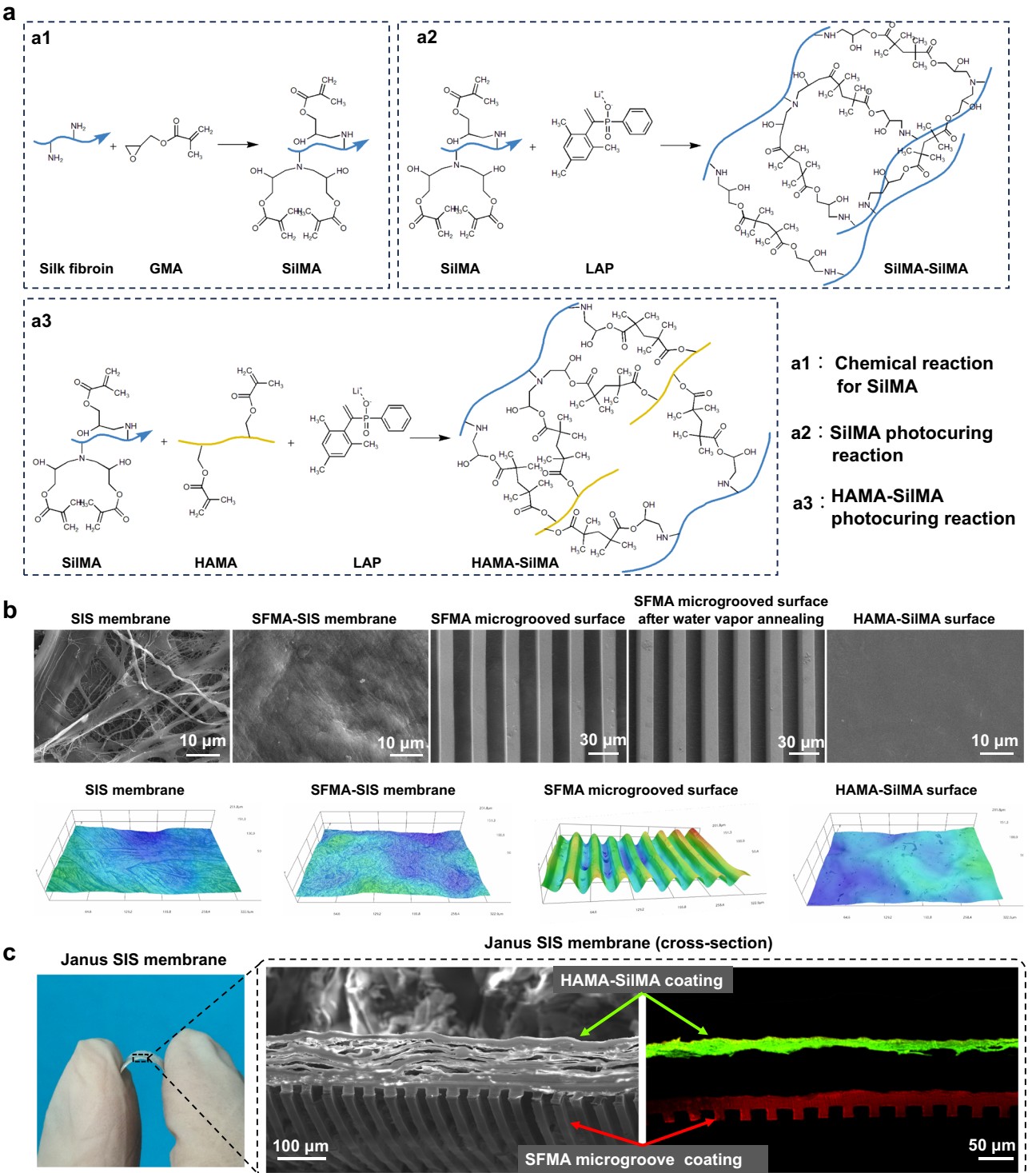

**Fig. 2 | Morphological characterization of Janus SIS membrane. a** SF chemically modified by glycidyl methacrylate (SilMA) and the photocuring reactions of SilMA and HAMA-SilMA. **b** SEM and 3D optical images of the SIS, SFMA-SIS, SFMA microgroove coating without water vapor annealing, SFMA microgroove coating after water vapor annealing, and HAMA-SilMA coating. **c** SEM and CLSM cross-section images of Janus SIS membrane. Red arrow indicates the SFMA microgroove coating; Green arrow indicates the HAMA-SilMA coating.

## The effects of water vapor annealing on physicochemical properties of the SFMA microgroove coating

Previous studies showed that the water vapor annealing had a significant impact on the physicochemical properties of SF[33]. Next, the secondary structure of SFMA microgroove coating was first investigated using Fourier transform infrared spectroscopy (FTIR). As shown in Fig. 3a, b, in the spectra of SFMA microgroove coating without and with 2 min water vapor annealing treatment, the peaks at 1644 cm$^{-1}$ and 1639 cm$^{-1}$ in the amide I region corresponded to random coils (1638−1655 cm$^{-1}$). Notably, the peak in the spectra of SFMA microgroove coating with 30 min water vapor annealing was identified at 1620 cm$^{-1}$, indicating its secondary structure was dominated by the β-

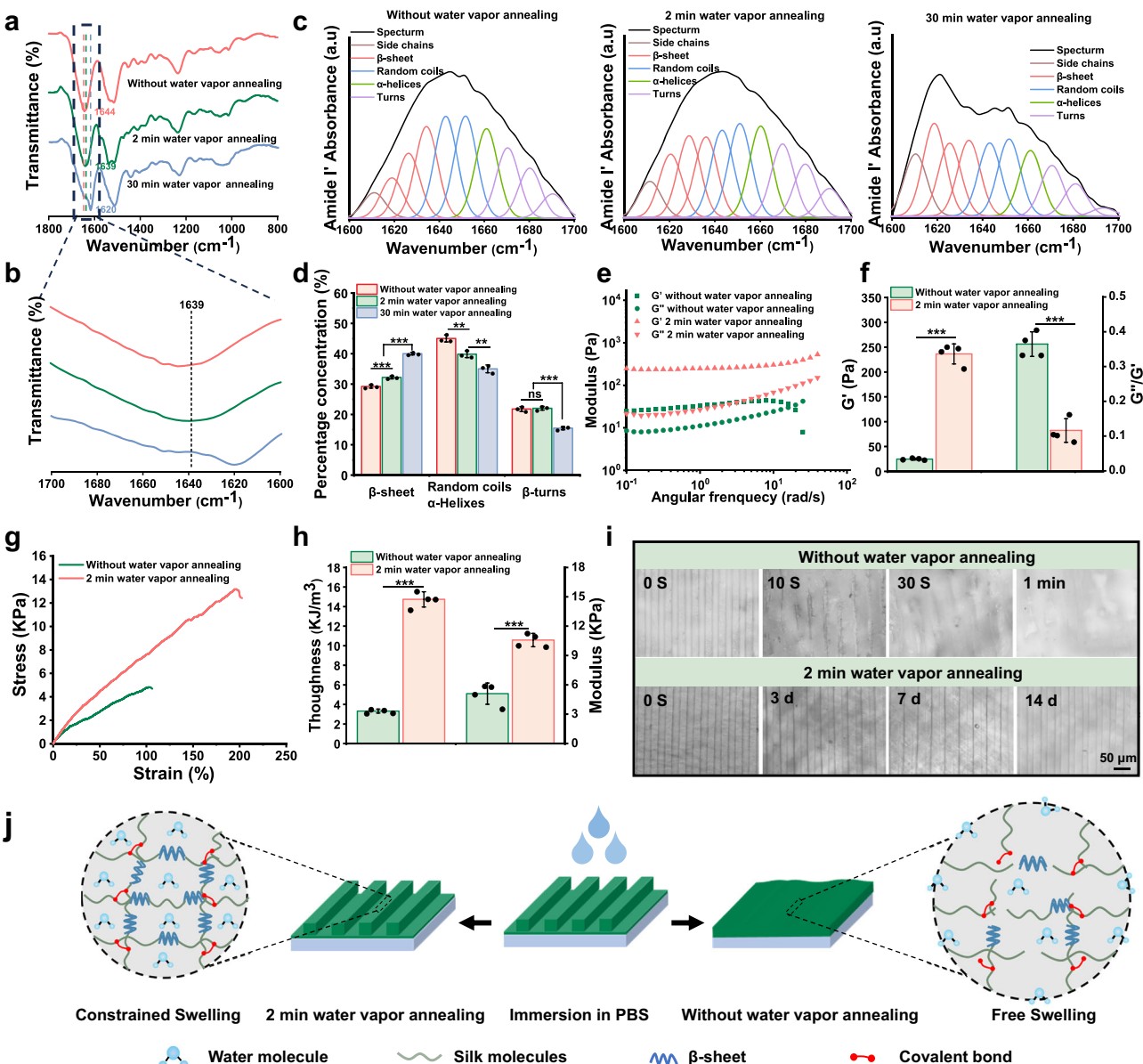

**Fig. 3 | Water vapor annealing enhanced the mechanical properties and structural stability of the SFMA microgroove coating. a** FTIR spectra of SFMA microgroove coating subjected to water vapor annealing treatment with different time. **b** Magnified view of the amide I region in (**a**) FTIR spectra. **c** Representative spectra at different water vapor annealing times after peak fitting. **d** Conformation contents in the amide I region of the SFMA microgroove coating at different water vapor annealing times. **e** Rheological curves of SFMA microgroove coating before and after 2 min water vapor annealing. **f** Quantitative analysis of storage modulus and loss angle of SFMA microgroove coating before and after 2 min water vapor annealing. **g** Tensile stress-strain curves of SFMA microgroove coating before and after 2 min water vapor annealing. **h** Quantitative analysis of toughness and

modulus of SFMA microgroove coating before and after 2 min water vapor annealing. **i** Optical images of SFMA microgroove coating without and after 2 min water vapor annealing immersed in PBS after different time lengths. **j** Schematic illustration of the anti-swelling mechanism of the SFMA microgroove coating without and after 2 min water vapor annealing. Values in (**d**, **f**, **h**) represent the mean ± SD (three independent replicates ($n = 3$) for d and four independent replicates ($n = 4$) for (**f** and **h**). Statistical difference was determined by two-tailed unpaired Student's $T$-test between two groups. One-way analysis of variance (ANOVA) with a Tukey's post hoc test was used for multiple comparisons. Source data and exact $P$ values are provided as a Source data file. (ns: $P > 0.05$, *$P < 0.05$, **$P < 0.01$, ***$P < 0.001$).

sheet (1616–1637 cm$^{-1}$). To further confirm the secondary structure content of the SFMA microgroove coating, the amide I region of the FTIR spectra was analyzed using Fourier self-deconvolution according to the established procedure[38]. It was found that the content of β-sheets increased with the extension of annealing time, and the random coil and α-helix decreased simultaneously (Fig. 3c, d). These results suggested that the water vapor annealing process promoted the formation of β-sheets, and the content of β-sheet could be controlled by the time of treatment.

After implantation in vivo, the SFMA microgroove coating could be rehydrated into a hydrogel state in a wet environment. Thus, the mechanical properties of the SFMA microgroove coating with and without water vapor annealing were systematically evaluated in rheological and tensile studies after complete hydration. The rheological test showed that the SFMA microgroove coating after water vapor annealing treatment exhibited a significant increase (≈tenfold) in storage modulus and a significantly lower loss angle (tan δ = G″/G′) compared to that without water vapor annealing at a fixed frequency

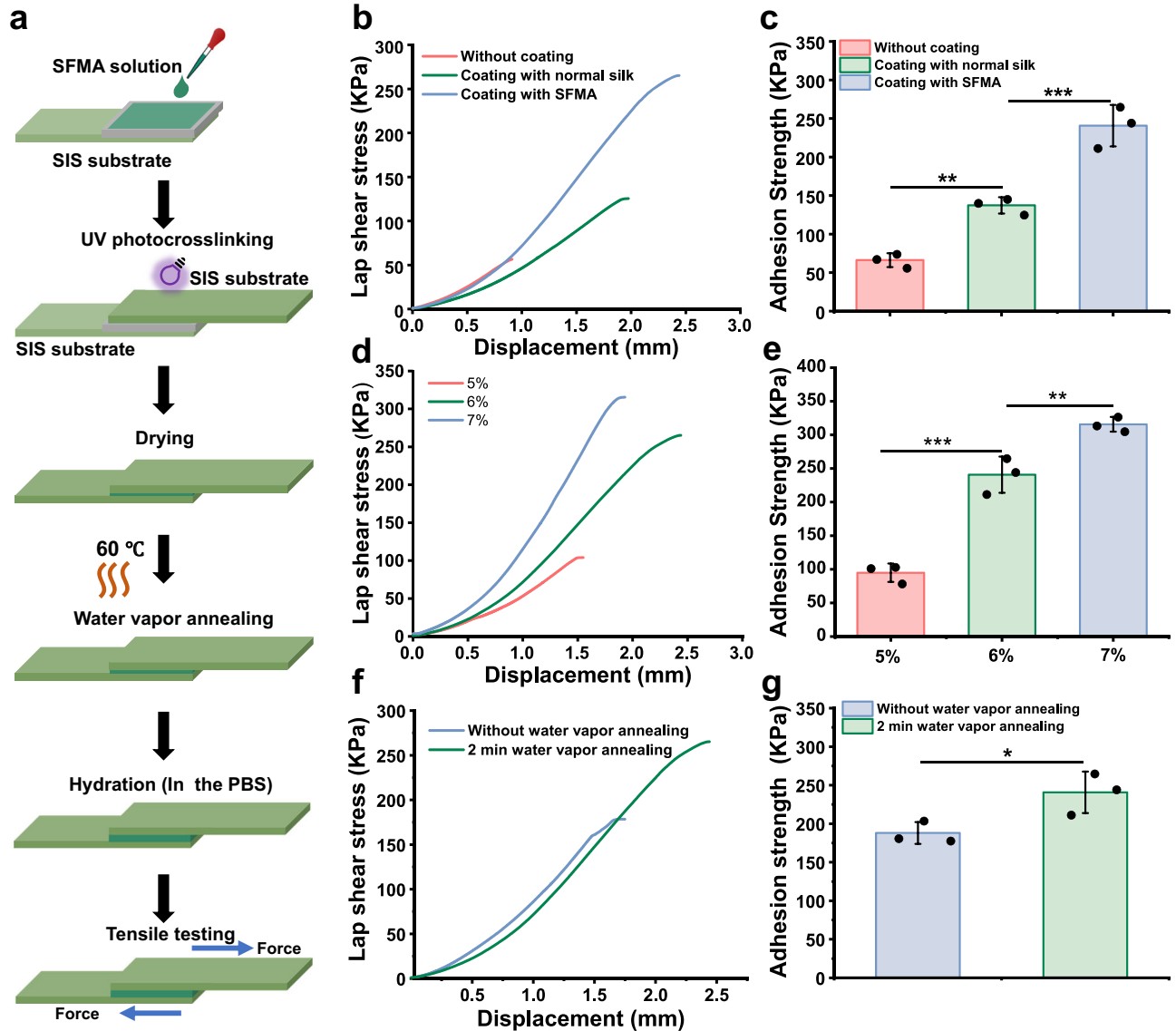

**Fig. 4 | Photocuring and water vapor annealing treatment enhanced the interfacial adhesion strength between the SFMA microgroove coating and the SIS substrate. a** Schematic diagram of the sample preparation and testing process for lap shear testing. **b, c** Lap-shear curves and adhesion strength analysis of SFMA microgroove coating on unmodified, normal silk-modified, and SFMA-modified SIS substrates. **d, e** Lap-shear curves and adhesion strength analysis of the SFMA microgroove coating with 5%, 6%, and 7% concentrations on SFMA-SIS substrates. **f, g** Lap-shear curves and adhesion strength analysis of the SFMA microgroove with or without water vapor annealing treatment on SFMA-SIS substrates. Values in (**c, e, g**) represent the mean ± SD (three independent replicates ($n = 3$)). Statistical difference was determined by two-tailed unpaired Student's T-test between two groups. One-way analysis of variance (ANOVA) with a Tukey's post hoc test was used for multiple comparisons. Source data and exact $P$ values are provided as a Source data file. (*$P < 0.05$, **$P < 0.01$, ***$P < 0.001$).

of 0.1 Hz (Fig. 3e, f). The tensile stress−strain results implied that the SFMA microgroove coating with water vapor annealing treatment exhibited higher tensile strength, break strain, Young's modulus and toughness than the coating without water vapor annealing (Fig. 3g, h), with a tensile strength of $12.69 \pm 2.06$ KPa (increased from $4.39 \pm 0.56$ KPa), break strain of 197% (increased from 132%), Young's modulus of $10.58 \pm 0.68$ KPa (increased from $5.1 \pm 1.98$ KPa), and toughness of $14.74 \pm 0.78$ KJ/m³ (increased from $3.32 \pm 0.2$ KJ/m³). These results reflected that water vapor annealing treatment could significantly enhance the mechanical properties of SFMA microgroove coating.

We further verified the structural stability of the SFMA microgroove coating on SFMA-SIS with and without water vapor annealing after immersion in phosphate buffered solution (PBS) for 3, 7, and 14 days. As shown, the SFMA microgroove coating treated with 2 min

water vapor annealing maintained their structural stability throughout the entire time, whereas untreated SFMA microgroove coating exhibited rapid swelling upon immersion in PBS solution, losing its clear microgroove structure completely within 1 min (Fig. 3i). We hypothesized that the increased β-sheet induced by water vapor annealing could act as crosslinking sites and strengthen the interactions between SF molecules, thereby enhancing the anti-swelling ability of the SFMA microgroove coating (Fig. 3j).

**The interfacial adhesion between the SFMA microgroove coating and SIS substrate**

Lap-shear testing was used to examine the interfacial adhesion between the SFMA microgroove coating and SIS substrate (Fig. 4a). The adhesion strength between the SFMA microgroove coating and SFMA-modified SIS substrate exhibited a strength of approximately

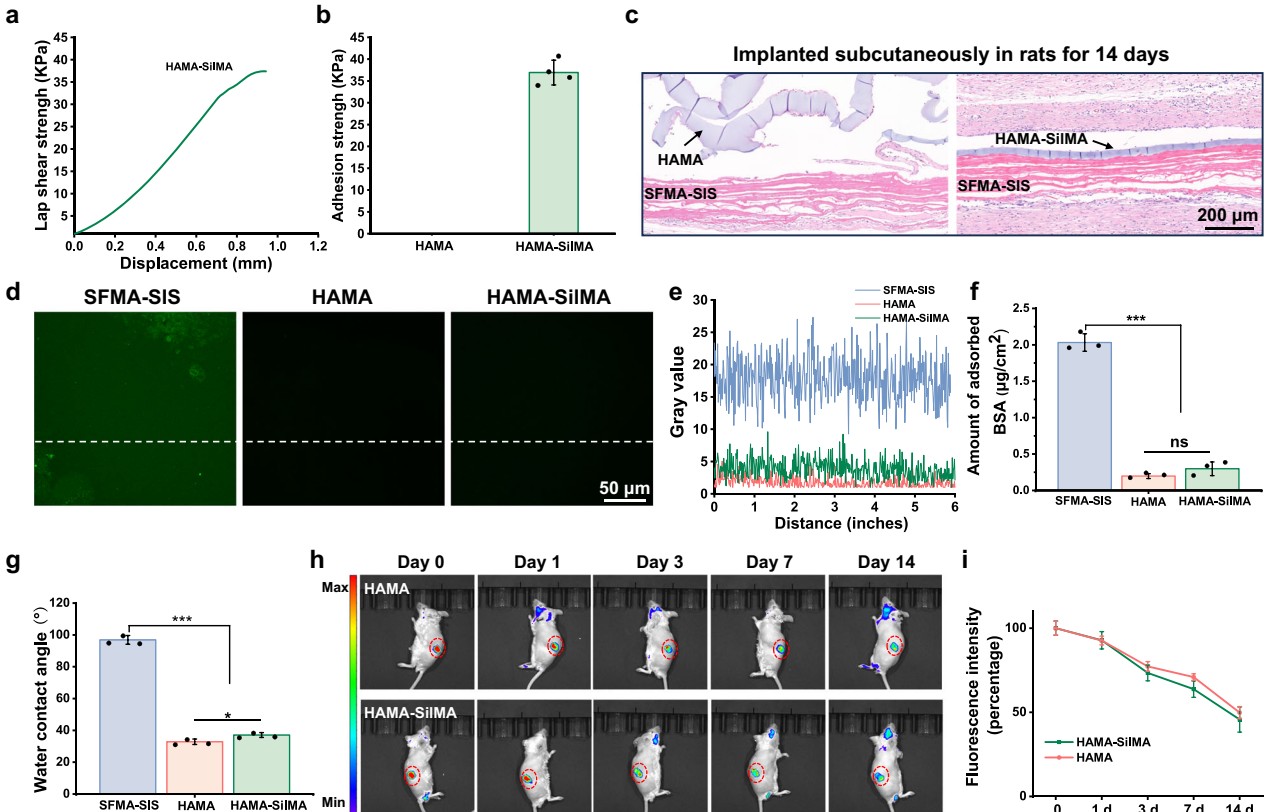

**Fig. 5 | The HAMA-SilMA coating demonstrated good stability on SFMA-SIS substrate and anti-adhesion performance. a, b** Lap-shear curves and adhesion strength analysis of HAMA and HAMA-SilMA coatings with SFMA-SIS substrates. **c** H&E images of HAMA- and HAMA-SilMA-coated SIS membranes implanted subcutaneously in rats for 14 days. **d** Fluorescence images of the nonspecific adsorption of FITC-labeled BSA on the SFMA-SIS, HAMA-coated SIS, and HAMA-SilMA-coated-SIS membranes. **e** Quantitative fluorescence intensity of (**d**) was obtained by line profiling (ImageJ). **f** Quantitative analysis of the adsorption protein content of different membranes. **g** Water contact angles of the SFMA-SIS membrane, HAMA coating, and HAMA-SilMA coating. **h** In vivo imaging of fluorescently labeled HAMA and HAMA-SilMA coatings subcutaneously implanted into BALB/c mice at various time points. **i** Quantitative analysis of fluorescence intensity in (**h**) at various time points. Values in (**b**, **f**, **g**, **i**) represent the mean ± SD (four independent replicates ($n = 4$) for b and three independent replicates ($n = 3$) for (**f**, **g** and **i**). Statistical difference was determined by two-tailed unpaired Student's $T$-test between two groups. One-way analysis of variance (ANOVA) with a Tukey's post hoc test was used for multiple comparisons. Source data and exact $P$ values are provided as a Source data file. (ns: $P > 0.05$, *$P < 0.05$, **$P < 0.01$, ***$P < 0.001$).

240.77 ± 26.95 KPa, which was significantly higher than that of normal silk-modified SIS (137.29 ± 10.57 KPa) and that of unmodified SIS (66.22 ± 9.04 KPa), probably due to the polymerization of double bonds of SilMA at the interface via the photocuring reaction (Fig. 4b, c). Meanwhile, as the concentration of SFMA was increased from 5% to 7% (w/v), the adhesion strength increased from 94.81 ± 13.69 KPa to 315.71 ± 11.02 KPa (Fig. 4d, e). We also evaluated the adhesion strength between the SFMA microgroove coating and SFMA-SIS with or without water vapor annealing (Fig. 4f, g). Impressively, the SFMA microgroove coating after water vapor annealing exhibited improved adhesion strength compared to that without treatment. Taken together, these results implied that the covalent bonding of the photocuring reaction and the physical interaction of β-sheets mediated by water vapor treatment synergistically increased the interfacial adhesion strength between the SFMA microgroove coating and the SIS substrate.

## Stability and anti-adhesion performances of the HAMA-based coating

HA and its derivatives have been widely used as anti-adhesion materials. However, HA has high water absorption capacity, which results in excessive swelling and insufficient endurance, limiting its application as a surface coating for implantable devices[39]. In this study, we added SilMA to a HAMA system to enhance the performance of the HAMA-based anti-adhesion coating. The FTIR spectroscopy results revealed a significant shift in the characteristic peaks of HAMA after the addition of SilMA, indicating effective HAMA-SilMA composite formation (Supplementary Fig. 5). To examine the effect of SilMA on the interface adhesion, lap shear test was performed. The results indicated that interface adhesion between the HAMA-SilMA coating and SFMA-SIS substrate was approximately 40 KPa. However, without the addition of SilMA, the adhesion strength between the HAMA coating and SFMA-SIS was hardly detectable (Fig. 5a, b). The results of rat subcutaneous implantation further demonstrated that after 14 days of implantation, the pure HAMA coating exhibited significant deformation and swelling and detached from the SFMA-SIS substrate, indicating that HAMA cannot maintain a stable structure in vivo in the long term. In contrast, the addition of SilMA effectively preserved the structure of the HAMA coating, preventing deformation, swelling, and detachment (Fig. 5c).

The adsorption of nonspecific proteins contributes to the formation of cell and tissue adhesion. To evaluate whether the addition of SilMA affects the anti-protein adsorption effect of the HAMA coating, we used fluorescently FITC-labeled bovine serum albumin (BSA) adsorption experiments to analyze the uncoated SIS membrane, SIS membrane with HAMA coatings or HAMA-SilMA coatings. The fluorescence intensity of CLSM images was observed and BCA protein was measured to determine the adsorption of proteins. We found that the fluorescence signals and amounts of BSA from the SIS membrane with

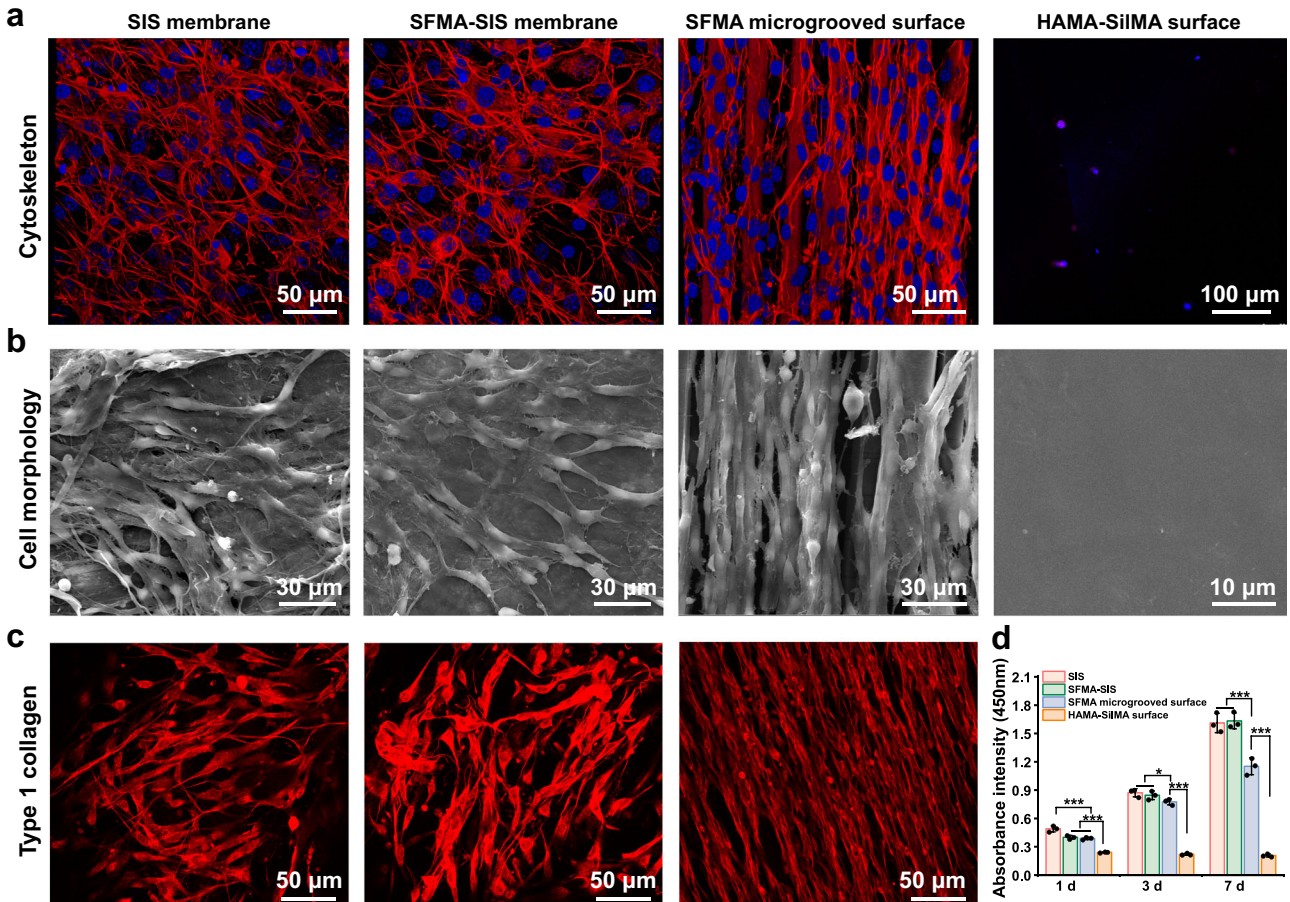

**Fig. 6 | The morphology, proliferation, and collagen I expression of fibroblasts cultured on silk-based coatings. a**, **b** CLSM and SEM images of the cytoskeleton and morphology of NIH3T3 fibroblasts on the unmodified SIS, SFMA-SIS, SFMA microgrooved surface, and HAMA-SilMA surface. **c** Immunofluorescence staining of collagen I secreted by fibroblasts on the unmodified SIS, SFMA-SIS, and SFMA microgrooved surfaces. **d** Cell proliferation analysis on different surfaces using the CCK-8 Kit. Values in (**d**) represent the mean ± SD (three independent replicates ($n$ = 3)). One-way analysis of variance (ANOVA) with a Tukey's post hoc test was used for multiple comparisons. Source data and exact $P$-values are provided as a source data file. (*$P$ < 0.05, **$P$ < 0.01, ***$P$ < 0.001).

HAMA coatings and HAMA-SilMA coatings were significantly lower than those of uncoated SIS, the amounts of BSA from the SIS membrane with HAMA coatings and HAMA-SilMA coatings were 9.71% and 14.66% that of uncoated SIS (Fig. 5d–f). Furthermore, the water contact angle test results revealed that the water contact angle decreased from 82.26.87 ± 1.47° to 31.87 ± 1.36°, and 23.95 ± 1.07° after being coated with HAMA-SilMA and HAMA, respectively, which significantly improved the hydrophilicity of the uncoated SIS membrane, as shown in Fig. 5g. These results indicated that the addition of SilMA did not significantly affect the hydrophilicity and anti-protein adsorption effect of HAMA.

For proper anti-adhesive efficacy, ideal anti-adhesion biomaterials need to have enough retention time, generally 2–4 weeks, to pass the whole stage of adhesion formation[40]. Here, we evaluated the in vivo degradation of the coatings by using an in vivo imaging system (IVIS) to measure the changes in fluorescence intensity of FITC-labeled HAMA and HAMA-SilMA coatings (Fig. 5h, i). The results showed that the fluorescence intensity of both coatings gradually decreased over time, and the fluorescence intensity of the HAMA and HAMA-SilMA coatings at 14 days of implantation was approximately 49.83% and 45.67% of the initial fluorescence intensity, respectively, indicating that the addition of SilMA did not significantly affect the degradation behavior of the HAMA-based coating, and the HAMA-based coating has appropriate degradation rate.

## In vitro biological function of silk-based coatings

To evaluate the impact of silk-based Janus coatings on cell behavior, NIH3T3 fibroblasts were seeded onto unmodified SIS, SFMA-SIS, SFMA microgrooved surface, and HAMA-SilMA surface. The cells on the SIS and SFMA-SIS exhibited a random arrangement after 3 days of culture. Due to the contact guidance by surface topology, cells on the SFMA microgrooved surface exhibited elongated shapes and grew in an organized manner (Fig. 6a, b). The CCK-8 test results showed that the cells on the SIS and SFMA-SIS groups had higher proliferation than those on the SFMA microgrooved surface after 1, 3, and 7 days of culture, and there was no significant difference between SIS and SFMA-SIS group (Fig. 6d). On the other hand, cells were barely observed on the HAMA-SilMA coating surface and showed very little proliferation. The HAMA-SilMA coating surface with no cell adhesion could contribute to the prevention of adjacent tissues from undesired contact.

Collagen, as a major component of the ECM, provides efficient mechanical and structural support to anisotropic tissues through its oriented arrangement and organized distribution. Hence, we used immunofluorescence staining to characterize the distribution of collagen I in fibroblasts cultured on the different substrates (Fig. 6c). We found that the distribution of collagen I fibers secreted by fibroblasts on the SIS and SFMA-SIS membranes is random, the collagen fibers exhibited an oriented arrangement on the SFMA microgrooves, revealing that the substrate topology has an important effect on the remodeling of the ECM.

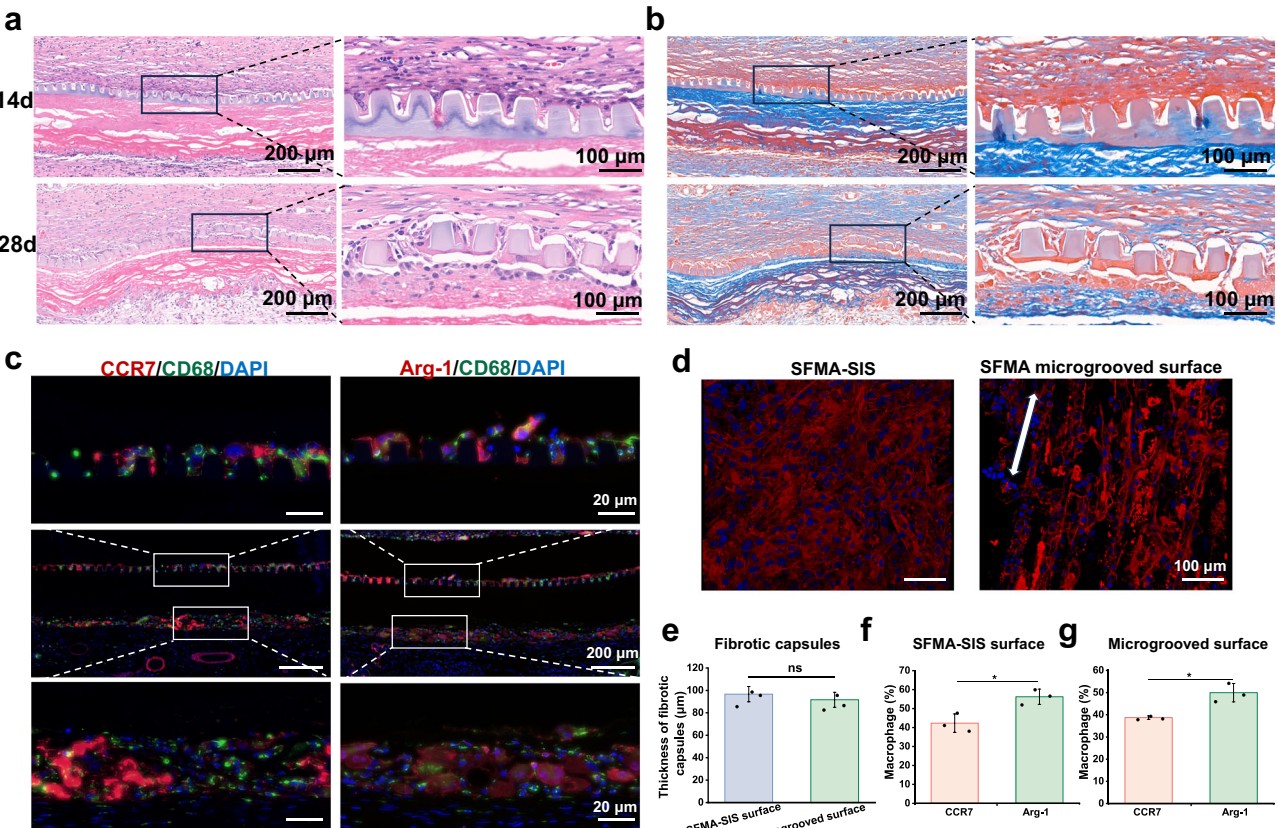

**Fig. 7 | SFMA microgroove coating integrated well with the host tissue and promoted M2 macrophage polarization. a**, **b** Representative H&E and Masson's trichrome staining images of SFMA microgroove coating after 14 and 28 days of subcutaneous implantation. **c** Immunofluorescence images of macrophages stained with the CD68, CCR7 and Arg-1 after 14 days of subcutaneous implantation. **d** Arrangement of host cells on the surface of the SFMA-SIS membrane and SFMA microgroove after 14 days of subcutaneous implantation. **e** Quantitative analysis of the thickness of fibrotic capsules after 14 days of subcutaneous implantation. **f**, **g** Quantitative analysis of CCR7⁺ and Arg-1⁺ macrophages in (**c**). Values in (**e**, **f** and **g**) represent the mean ± SD (three independent replicates ($n = 3$)). Statistical difference was determined by two-tailed unpaired Student's *T*-test between two groups. Source data and exact *P*-values are provided as a source data file. (ns: $P > 0.05$, *$P < 0.05$, **$P < 0.01$, ***$P < 0.001$).

## In vivo histological assessment of SFMA microgroove coating

To further evaluate the biological responses of the silk-based coatings in vivo, we implanted different membranes subcutaneously in SD rats for 28 days. Hematoxylin and eosin (H&E) and Masson's trichrome staining were performed to analyze the in vivo microgroove stability and tissue integration. As presented in Fig. 7a, b, the results revealed that the SFMA microgroove coating adhered tightly to the SFMA-SIS membrane and maintained the intact microgroove structure on day 14 of implantation. After 28 days of implantation, the structure of the microgroove had disintegrated, indicating that the SFMA microgroove had undergone biodegradation. Additionally, the large number of host cells distributed inside the microgroove, and the SFMA microgrooved surface and SFMA-SIS surface showed similar fibrous capsule formation (Fig. 7e), suggesting the well integration of the microgroove coating with the host tissue.

Macrophages play a pivotal role in tissue repair due to their plasticity and direct function in the immunomodulation. A switch to the M2 phenotype has been shown to be beneficial for tissue healing. Therefore, we further analyzed macrophage phenotype at the interface between SFMA microgroove coating and host tissue. Immunofluorescence stainings of pan-macrophage (CD68), M1 (CCR7), and M2 (Arg-1) phenotypes were performed after 14 days of implantation (Fig. 7c, f, g). The results showed predominantly Arg-1-positive macrophages on both the SFMA microgrooved surface and the SFMA-SIS surface, suggesting that similar to bioactive SIS membranes, SFMA microgroove coating promote the polarization of macrophages toward the M2 phenotype. We also observed host cells on the SFMA microgrooved surface for 14 days after implantation using cytoskeleton staining and found that the host cells showed directional distribution (Fig. 7d). This phenomenon is consistent with the results of the in vitro cell assessment.

## In vivo histological evaluation of HAMA-SilMA coating

Histological evaluation revealed that the HAMA-SilMA coating surface acted as a protective barrier was clearly separated from the host tissue and effectively prevent the invasion of host cells and tissue after 14 days implantation (Fig. 8a). On the contrary, the SFMA-SIS membrane side showed good tissue integration accompanied by host cells infiltration. This result indicated that the HAMA-SilMA coating had an excellent isolation effect and could prevent the ingrowth of host cells and tissues, thereby reducing the degree of tissue adhesion. Subsequently, we analyzed the macrophage polarization, collagenous fibrotic capsules and myofibroblasts expression to determine the degree of fibrosis. According to Fig. 8b–f, macrophages on the HAMA-SilMA side presented a higher M1 phenotype, however, the results showed that the HAMA-SilMA side exhibited a thinner fibrous capsule and lower expression of α-SMA-positive fibroblasts, compared to the SFMA-SIS side. Collectively, these results indicated that the HAMA-SilMA coating could effectively mitigate fibrotic formation.

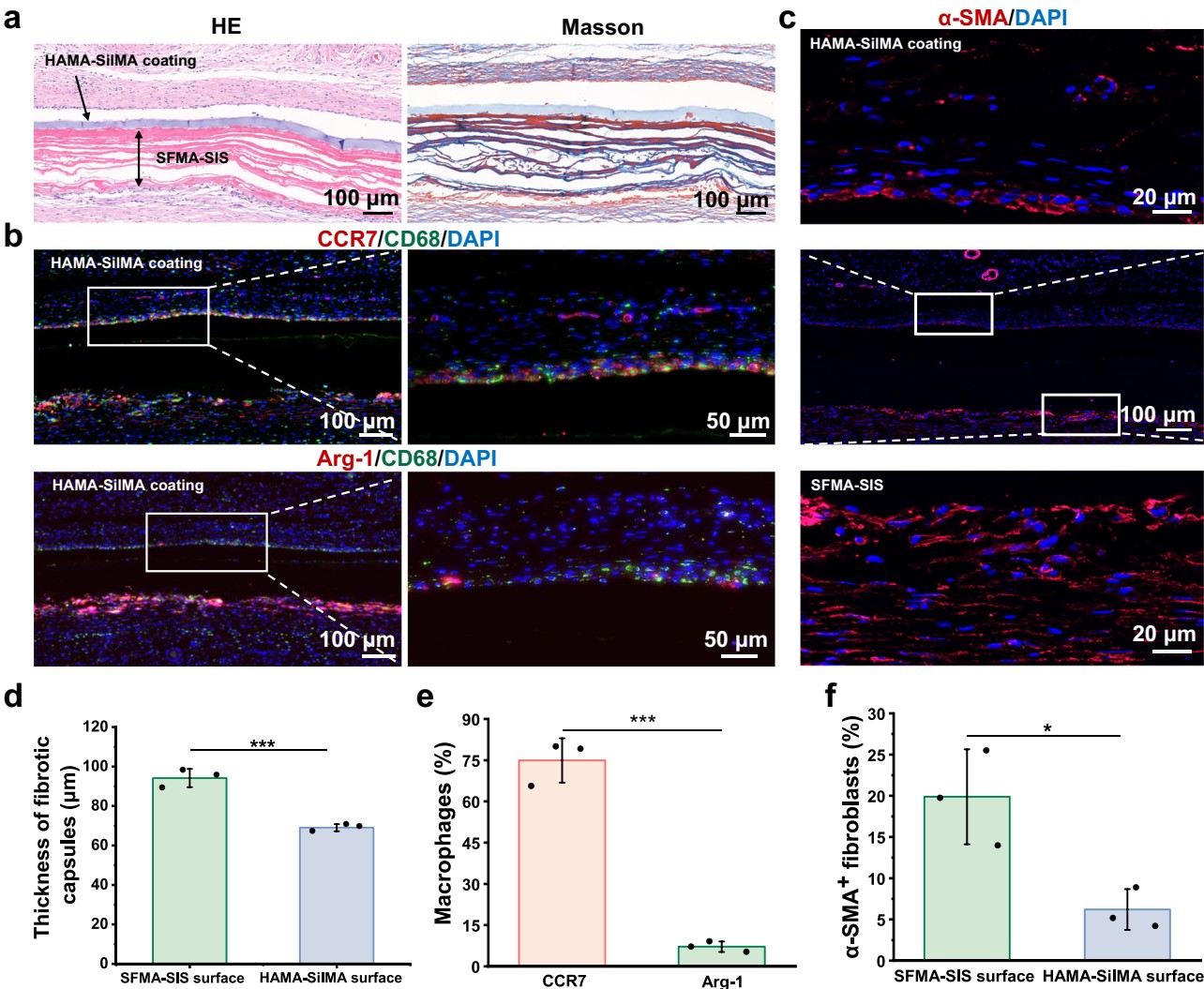

**Fig. 8 | HAMA-SilMA coating inhibited tissue ingrowth and mitigated fibrosis formation. a** Representative H&E and Masson's trichrome staining images of HAMA-SilMA coating after 14 days of subcutaneous implantation. **b** Immunofluorescence images of macrophages stained with the CD68, CCR7, and Arg-1 after 14 days of subcutaneous implantation. **c** Immunofluorescence images of fibroblasts stained with the α-SMA after 14 days of subcutaneous implantation. **d** Quantitative analysis of the thickness of fibrotic capsules after 14 days of

subcutaneous implantation. **e** Quantitative analysis of the Arg-1+ and CCR7+ macrophages of HAMA-SilMA coating after 14 days of subcutaneous implantation. **f** Quantitative analysis of the α-SMA+ fibroblasts after 14 days of subcutaneous implantation. Statistical difference was determined by two-tailed unpaired Student's T-test between two groups. Values in (**d**, **e** and **f**) represent the mean ± SD. (three independent replicates (n = 3)). Source data and exact P-values are provided as a source data file. (*P < 0.05, **P < 0.01, ***P < 0.001).

## Janus SIS for repairing spinal dural defects in rats

In this study, we used a spinal laminectomy model for creating a dural defect to characterize the therapeutic performance of Janus SIS membrane in vivo. The detail of surgical procedure was demonstrated in Supplementary Fig. 8 and Supplementary Movie 1. The experimental design was shown in Fig. 9a. The spinal tissue macroscopic images showed that compared with the white smooth surface of the natural dura mater, the injury group (control) without material treatment was found to have a wrinkled morphology and a brown color. A small amount of brown tissue also appeared on the dura mater surface of the SIS membrane repair group, while the Janus SIS repair group showed similar symptoms to the natural dura mater (Fig. 9b). H&E and Masson's trichrome staining showed that SIS group presented irregular and discontinuous newly formed collagen tissue observed in defect areas. Continuous newborn collagen tissue was observed in the dura defect area implanted with Janus SIS group, showing high similarity to normal dura tissue. Moreover, the collagen tissue integrated well with the normal dura tissue. On the contrary, there was almost none of newborn collagen

tissue in the control group without materials implantation due to the lack of regeneration ability of spinal dura mater (Fig. 9c, d, f). Magnetic resonance imaging (MRI) was used to observe the degree of scar adhesion and fibrosis of epidural tissue at 8 weeks after operation. As shown in Supplementary Fig. 9, in the control group, severe scar fibrosis in the surgical area could be observed on the MRI images, and the local scar fibrous tissue was found to immerse in the spinal cord. In the SIS and Janus SIS group, the MRI images exhibited a relatively low signal between the spinal cord and the surrounding tissue, indicating the limited fibrosis scar formation, and compared with SIS group, the Janus SIS showed a clearer boundary between spinal cord and the epidural tissue. We further performed immunofluorescence staining of α-SMA and collagen I to assess epidural fibrosis adhesion (Fig. 9e, g, h). The results showed that a large number of α-SMA-positive cells were clustered at the epidural tissue in the control group, and the number of α-SMA-positive fibroblasts was significantly lower in the SIS and Janus SIS groups. Additionally, the Janus SIS group showed fewer α-SMA-positive cells within the epidural tissue compared to the SIS group.

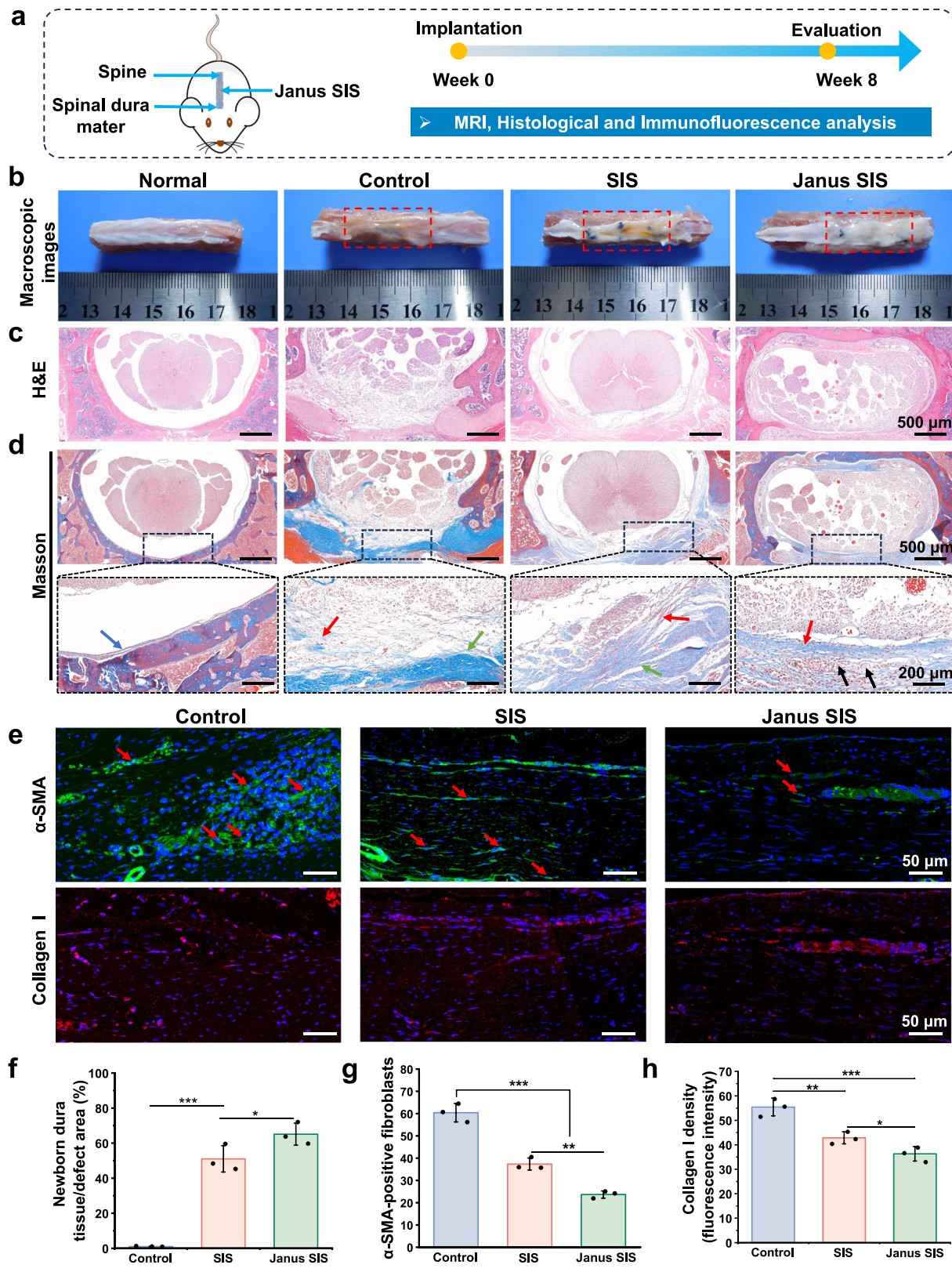

In addition, extensively dense collagen I occupied the epidural area in the control group compared with the SIS and Janus SIS groups, while significantly lower collagen I expression could be found in groups implanted with SIS and Janus SIS. These promising results suggested that our Janus SIS membrane could lead to more effective repair of spinal dura mater defects and provide good application prospects for clinical treatment of the spinal dura mater.

## Discussion

The specific microenvironment in spinal dura mater put forward multiple requirements on designs of regenerative materials, which should be capable of simultaneously guiding tissue cells directional growth and inhibiting epidural fibrosis adhesion. The realization of pro-regeneration and anti-adhesion is based on contradictory cells adhesion and anti-adhesion behaviors, which requires the successful

**Fig. 9 | In situ dural mater regeneration enhanced by Janus SIS in a rat spinal dural defect model. a** Experimental design include MRI, histological analysis and immunology evaluations. **b** Macroscopic photographs of spinal tissues at 8 weeks after operation, including the normal control (without injury), control group, SIS group and Janus SIS group. **c, d** Histological evaluation (H&E and Masson's tri-chrome staining) of spinal dural mater repair in different groups. The blue arrow indicates the normal spinal dura mater. The red arrow indicates newborn dura mater-like tissue. The green arrow indicates fibrotic tissue. The black arrow indicates residual SFMA microgroove coating. **e** Immunohistochemical staining of α-SMA and collagen I at the focal area of epidural tissues in different groups. **f** Quantitative analysis of newborn collagen tissue in the defect area. **g, h** Quantitative analysis of α-SMA positive cells and collagen I expression in histological sections. One-way analysis of variance (ANOVA) with a Tukey's post hoc test was used for multiple comparisons. Values in (**f, g**, and **h**) represent the mean ± SD (three independent replicates ($n = 3$)). Source data and exact $P$-values are provided as a source data file. (*$P < 0.05$, **$P < 0.01$, ***$P < 0.001$).

and simultaneous management of conflicting properties[41,42]. Recently, the fabrication of Janus-type material that with distinct functions on the two surfaces exhibited indeed successful in promoting spinal dura mater repair[1]. Therefore, more efforts should be made to develop a biomimetic Janus material to promote regeneration and reconstruction of the spinal dura mater.

Due to the natural components carrying bioactive macromolecules, SIS exhibits excellent biological functions in promoting tissue regeneration[43,44]. In this study, we constructed a Janus SIS with ECM-biomimetic anisotropic structure and anti-adhesion hydrogel coatings that could satisfy the multiple functional demands of spinal dura mater repair. Increasing evidences have demonstrated that anisotropic structures affect the remodeling of the ECM by modulating fibroblast adhesion status and morphology, promoting tissue healing[45]. Meanwhile, Hydrogels are water-swollen networks and have similarity to native ECM microenvironment. For simulation of natural spinal dura mater ECM, here, we prepared a microgroove hydrogel coating by using the SF and SilMA blends. The higher crystallinity and MW of SF endows it with stronger structure and mechanical stability. However, higher crystallinity makes it resist to water absorption. The SilMA with lower MW and crystallinity has the ability of in-situ modeling and hydration ability (Supplementary Figs. 2, 3 and 6). In addition, our developed process involving water vapor treatment at 60 °C guided SF to form β-sheet structures. Compared with traditional methanol treatment, water vapor annealing is a relatively green approach that can effectively mediate the formation of SF β-sheet structure[33]. Here, we selected 2 min treatment time, which not only improved the mechanical strength and resistance to swelling of the coating, but also enabled hydration, however, longer annealing time could not achieve rehydration (Fig. 3 and Supplementary Fig. 7). The combination of SF with different MW and crystallinity with the 2 min water vapor treatment, realizing the fabrication of SFMA microgroove hydrogel coating with good anti-swelling ability and structural stability. Moreover, studies have shown that cells can perceive structural cues that are approximately the same size as the cells themselves[46]. Therefore, we used 20 μm PDMS microgroove templates to fabricate SFMA microgroove hydrogel coating.

To obtain anti-tissue adhesion surface, we utilized HAMA as a hydrophilic macromolecular coating to reduce nonspecific protein adsorption and tissue-material integration. The rationale for the use of HA in the prevention of adhesion formation is based on several biological effects, including barrier effect to fibrinogen and cells, modulation of fibrous tissue formation[26]. However, due to HA's tendency to swell and its relatively weak structural and mechanical stability in vivo, its long-term anti-tissue adhesion effectiveness is limited[47]. In this study, the introduction of SilMA significantly reduced HAMA's in vivo swelling and enhanced its long-term stability in interfacial bonding with the SIS substrate (Fig. 5a, b, c). Notably, the addition of SilMA did not alter the hydrophilic properties, anti-protein adsorption capabilities, or degradation of HAMA (Fig. 5d–i).

In subcutaneous implantation experiments in rats, we found that host cells could enter the microgrooves after implantation, enhancing the integration between the tissue and the material (Fig. 7a, b). Studies have shown that the anisotropic surface could influence immune reactions of macrophages[48,49]. In present study, immunofluorescence analysis showed that the SFMA microgroove coating significantly increased the proportion of M2 macrophages (Fig. 7c, f, g), consistent with previous classic research showing that the 20 μm size microgroove, which are appropriately sized relative to cells, could effectively promote macrophages to express more Arg-1 proteins[50]. A high proportion of M2 macrophages promotes tissue repair and regeneration, aligning with the requirements for guiding spinal dura mater repair. Furthermore, the HAMA-SilMA surface effectively reduced the infiltration and integration of host cells and tissues, thereby increasing physical isolation. Although a high proportion of M1 macrophages appeared at the interface, there was thinner fibrous capsules and lower accumulation of myofibroblasts, it may be attributed to the biological effects of HA[11,26,27]. Importantly, in the rat spinal dura mater injury repair model, the Janus SIS membrane exhibited tissue regeneration similar to the dura mater structure. Both the SIS and the Janus SIS underwent complete degradation over 2 months of in vivo implantation. Meanwhile, there were residual SFMA microgroove, indicating the positive role of slowly degrading SFMA microgroove hydrogel coating in dura mater repair. Immunofluorescence results also revealed that the HAMA-SilMA coating could significantly suppress the adhesion and proliferation of myofibroblasts, thus inhibiting the development of epidural fibrosis adhesion.

## Methods

We have complied with all relevant ethical regulations declared in the manuscript. All animal procedures were approved by the Biomedical Ethics Committee of Beihang University (Number: BM20200180). Animals were housed on a 12/12-h light/dark cycle, 21–25 °C, 40–70% humidity with adequate food and water.

### Materials

SilMA (6-10KD), HAMA (400KD), LAP and vinyl coupled fluorescent dyes (DYE-UF-ENE-R/G) were purchased from Engineering for Life (EFL, Suzhou, China). Polyethylene glycol (PEG, Mn-100,000 g/mol), sodium bicarbonate ($NaHCO_3$, 99.8%) and cyclohexyl isocyanide (≥98%) were purchased from Macklin Chemical Reagent Co., Ltd., China. Lithium bromide (LiBr, 99%), fluorescein amine isomer I (≥90%) and acetaldehyde (≥99.5%) were purchased from Sigma-Aldrich.

SIS was prepared as follows: porcine small intestines were harvested from healthy home-raised pigs, then cut into segments and washed with a saline solution. The SIS was obtained by mechanically removing the tunica serosa and tunica muscularis, followed by soaking the submucous membrane in a solution containing methanol and chloroform (1:1, V/V) for 12 h, and then incubated in 0.05% trypsin for 12 hours. The membrane was further treated with 0.5% sodium dodecyl sulfate (SDS), and then soaked in 0.1% peroxyacetic acid. After each treatment above mentioned, the membranes were thoroughly rinsed with saline solution. Finally, the membranes were freeze-dried and sterilized using gamma irradiation[51].

SF solution was extracted from raw silk (Zhejiang Xingyue Biological Technology Co., Ltd, China). In brief, raw silk (10 $g$) were degummed for 1 h in a boiling $NaHCO_3$ solution (2 L, 5 g/L), and sericin was removed with water. The degummed silk fibers were dried at 45 °C for 24 h. Subsequently, the dried degummed silk fibers were dissolved in LiBr (100 mL, 9.3 M) at 40 °C for 3 h. The SF solution was then dialyzed with deionized water (MWCO 8000–14000) for 3 days to remove ions and impurities, the deionized water was changed 3 times a day.

The purified SF solution (~20 mg/mL) was obtained. The highly concentrated SF solution (60 mg/mL) was obtained by using a dialysis membrane (MWCO 3500) in a 15% PEG solution[52].

Fluorescent HAMA was prepared as follows: 50 mg HAMA was completely dissolved in 40 ml of deionized water and fully mixed with 15 ml of DMSO. 25 mg fluorescein amine isomer I was dissolved in 5 ml of DMSO, followed by the addition of 25 μl of acetaldehyde and 25 μl of cyclohexyl isocyanide. The solution was then added to the prepared HAMA solution and stirred at room temperature for 5 h. The resulting solution was dialyzed (MWCO 2000 Da) to remove unreacted fluorescein amine molecules and other reagents. Finally, the fluorescently labeled HAMA (f-HAMA) solution was dried in an oven at 40 °C and further freeze-dried to completely remove the residual water.

## Fabrication of Janus SIS

To fabricate Janus SIS, the 1 mg/ml SFMA solution was first deposited on the SIS surface via LbL self-assembly. Briefly, the SIS membranes were incubated with 1 mg/ml SFMA solution at 4 °C for 15 min and washed 3 times with ultrapure water. Then, the SFMA-coated SIS (named SFMA-SIS) were immersed in 90% methanol for 15 min. The SFMA-SIS were dried under nitrogen gas and subjected to the next coating procedure[36]. Subsequently, the PDMS mold with a 20 μm microgroove structure was treated with plasma to endow it with a hydrophilic surface. Then, a 15 mm × 15 mm silicone frame was placed on it, followed by the addition of 60 mg/ml SFMA solution (LAP 0.25%). The SFMA-SIS was covered on the solution followed by photocuring with UV irradiation (405 nm) for 35 s at the density of 25 mW/cm² to generate the corresponding microgrooves. Next, on the top surface of the SFMA-SIS, SilMA (5 mg/ml, LAP 0.25%) and HAMA (15 mg/ml, LAP 0.25%) mixture was smeared uniformly and photo-polymerized under UV light (405 nm, 25 mW/cm²) for 15 s and then dried at 37 °C. Finally, the Janus SIS was stabilized by 60 °C water vapor annealing for 2 min.

## Characterization of Janus SIS

The surface and cross section morphology of the Janus SIS was first characterized by SEM (S-4800, Hitachi, Japan) after the membrane was sputter-coated with gold for 120 s. Subsequently, CLSM (Leica SP8, Germany) was used to image the DYE-UF-ENE-G-labeled HAMA-SilMA coating and DYE-UF-ENE-R-labeled SFMA microgroove coating at excitation wavelengths of 488 and 543 nm. The surface chemical and conformation analysis of the coating was carried out on an FTIR-7600 spectrometer (Lambda Scientific, Australia) with a wavenumber range of 600–4000 cm⁻¹. The peak fit 4.0 was used to deconvolute the conformation in the amide I region (1600–1700 cm⁻¹). The hydrophilicity was tested using a contact angle system (Shanghai Powereach Digital Technology Equipment Co. Ltd, China). Two microliters distilled water was dropped onto the membrane surface, an image of the water droplet was captured at the same time scale, and the water contact angle was calculated.

## Mechanical testing of the SFMA coating

To characterize the effect of the water vapor annealing treatment process on the mechanical properties of the SFMA microgroove coating, a rheological test was first conducted. A round SFMA coating sample with a diameter of 25 mm was tested on a DHR-2 rheometer (TA instruments, Waters Ltd.). Frequency scanning was performed in the range of 0.1–100 Hz with a constant oscillating strain of 5% to obtain the storage modulus (G′) and loss modulus (G″) of the sample. All experiments were carried out at 25 °C.

The tensile properties of the SFMA microgroove coating were measured using a dynamic mechanical analyzer (DMA Q800, TA Instruments, Waters Ltd, USA) at a speed of 0.5 mm/min at room temperature. The resulting mechanical parameters, including tensile strength, fracture strain, modulus and toughness were determined.

## SFMA microgroove coating structure stability

The anti-swelling capability of the SFMA microgroove coating was evaluated by immersing the sample in PBS (Solarbio, China) solution at room temperature. The morphology changes of the SFMA microgroove coating with or without water vapor annealing at designated points during hydration tests were recorded by microscope.

## Interface adhesion performance

The interface adhesion strength between the SFMA microgroove coating and SIS substrate, HAMA-SilMA coating and SIS substrate were measured by the lap-shear test according to ASTM F2255 using a dynamic mechanical analyzer (DMA Q800, TA Instruments, Waters Ltd, USA). Briefly, a 20 mm × 20 mm wet SIS substrate was laid flat on a glass sheet, and then a silicone frame with an inner diameter of 20 mm × 5 mm was placed on one end of the SIS substrate. Next, the SFMA or HAMA-SilMA solution was dropped into the frame and then covered with another 20 mm × 20 mm of the same SIS substrate on top of the solution followed by UV irradiation for 35 or 15 s. After dried at 37 °C, the samples were cut to a width of 2 mm, followed by water vapor annealing treatment and hydrated in PBS for 15 min before testing. The adhesion strength was determined as the maximum tensile force ($F_{max}$) per contact area.

## Protein adsorption measurement

Fluorescently labeled BSA was selected as a model biofouling protein for the nonspecific adsorption test. Samples of SFMA-SIS, HAMA-SilMA coated SIS and HAMA coated SIS were first incubated in PBS for 2 h. Then, the samples were incubated in 1 mg/ml FITC-labeled BSA (Solarbio, China) at 37 °C for 2 h. Afterward, the samples were thoroughly rinsed and observed using CLSM (Leica SP8, Germany). The quantitative fluorescence intensity was obtained by ImageJ software. To quantitatively characterize the amount of adsorbed protein, the samples were further treated with 1% SDS (Solarbio, China) to remove the adsorbed protein for 1 h. The amount of adsorbed protein was quantified using a BCA protein assay kit (Solarbio, China).

## Degradation test

The in vivo degradation behavior of the HAMA-SilMA coating SIS and HAMA coating SIS (as control) was monitored by a small animal optical IVIS (PerkinElmer, USA). Three 6–8 weeks male BALB/c mice were raised in the mouse feeding system in accordance with the guiding principles for animal care and use of Beihang University and were approved by the Biomedical Ethics Committee of the Beihang university (Number: BM20200180). After sterilization, a small midline incision was made on the dorsum of each mouse, and the f-HAMA-SilMA and f-HAMA coating SIS were implanted into the left and right sides of the incision. After implantation, the mice were anesthetized with isoflurane (2% in oxygen) during imaging. Imaging was performed according to the manufacturer's instructions.

## Cell experiment

Mouse embryonic fibroblasts (NIH3T3 fibroblasts, GNM 6) were obtained from the Cell Bank of the Chinese Academy of Science. NIH3T3 cells (1 × 10⁵/mL) were cultured in Dulbecco's modified Eagle's medium (DMEM, high glucose, Invitrogen, USA) supplemented with 10% fetal bovine serum (FBS Gibco, USA) and 1% streptomycin/penicillin (Gibco, USA). The culture dishes were placed in a 5% CO₂ incubator at 37 °C. The culture medium was replaced every two days.

CLSM and SEM were used to observe the cellular spreading on the membranes. Briefly, after the cells were cultured on the membranes for 3 days, the cellular samples were fixed with 4% paraformaldehyde. For CLSM, after rinsed three times with PBS, samples were treated with 0.1% Triton X-100 and further blocked with 2.5% BSA. Staining was conducted using TRITC phalloidin (1:200 dilutions, Solarbio, China)

and mounting medium with DAPI (Solarbio, China). For SEM, cellular samples were dehydrated with graded concentrations (30, 50, 70, 90, 100% v/v) of ethanol, and subsequently, the samples were coated with gold for the observation of cell morphology.

Cell proliferation was measured using cell counting kit-8 reagent (CCK-8, Dojindo, Kumamoto, Japan). After 1, 3, and 7 days of culture, the culture medium was removed, and fresh medium containing 10% CCK-8 solution was added and incubated with cells at 37 °C for 1 h. Subsequently, the supernatant was transported into new 96-well plates to determine the absorbance at 450 nm using a spectrophotometer (Thermo Scientific, USA).

The collagen matrix secreted by NIH3T3 cells cultured on the SIS, SFMA-SIS and SFMA microgroove coatings was detected by collagen I immunofluorescence staining. Briefly, after the cells were cultured on the membranes for 3 days, the cellular samples were fixed with 4% paraformaldehyde for 30 min, followed by washing in PBS three times. Then, the samples were treated with 0.1% Triton X-100 for 5 min at room temperature. After blocked with 2.5% BSA, samples were stained with a primary anti-collagen I monoclonal antibody (1:200, BOSTER, China) at 4 °C overnight, followed by incubation with secondary antibody (Alexa Fluor 594 labeled IgG, Solarbio, China) at 37 °C for 1 h. Finally, mounting medium with DAPI was used to cover the slides for CLSM observation.

### Subcutaneous implantation

All animal procedures were approved by the Biomedical Ethics Committee of Beihang University (Number: BM20200180). For the evaluation of histocompatibility, six adult male SD rats (6–8 weeks) were used for subcutaneous implantation experiments to evaluate the compatibility of the coated-SIS in vivo. After anesthesia, a small midline incision was made on the dorsum of rats, and the SIS membranes with SFMA microgroove coating and HAMA-SilMA coating were implanted into bilateral subcutaneous pockets of the incision. The rats were sacrificed at 14 and 28 days after the operation, and the implanted membranes were removed with the surrounding tissue. The samples were fixed and processed for the following histological analysis. For the observation of host cells on the SFMA microgroove coating, the SIS membrane with the SFMA microgroove coating was implanted into the subcutaneous pockets on the dorsum of three rats. The rats were sacrificed at 14 days after the operation, and the implanted membranes were removed and rinsed with PBS, then the membranes were fixed and the cytoskeleton staining was performed.

### Spinal dura mater defect model

All animal procedures were approved by the Biomedical Ethics Committee of Beihang University (Number: BM20200180). For spinal dura mater defect repair of Janus SIS, a rat duraplasty model was prepared[1]. A total of sixteen male SD rats (6–8 weeks) were divided into four groups: normal control: without defect modeling, control group: defect modeling without membrane implantation, SIS group: defect modeling with SIS implantation, and Janus SIS group: defect modeling with Janus SIS implantation. After anesthesia, the skin, subcutaneous tissue and deep fascia of rats were cut, and the paraspinal muscles on both sides were stripped. Then, the lamina between L4 and L5 was partial removed to expose the spinal dura mater. A dural defect about 3 mm × 8 mm in size was created. Eight weeks after the operation, the spinal tissues of the surgical segment were harvested. The samples were fixed and processed for subsequent histological analysis.

### Histology and immunofluorescence staining

For histological analysis, cross-sections of the spine samples with thickness of 5 µm are made for H&E staining, Masson's trichrome staining and immunological evaluations. H&E and Masson's trichrome staining were performed to assess the stability of the coating, tissue integration, anti-adhesion performance, collagenous fibrotic capsule formation, and regeneration of the defect dura mater. For immunofluorescence staining, all sections were incubated with primary antibodies against CD68 (1:200, ab31630, Abcam, USA), CCR7 (1:200, ab32527, Abcam, USA), Arg-1 (1:200, ab91279, Abcam, USA), α-SMA (1:100, ab32575, Abcam, USA) and Collagen I (1:50, BA0325, BOSTER, China) at 4 °C overnight, followed by treatment with Alexa Fluor 488/594-labeled secondary antibody. Finally, the samples were rinsed with PBS and stained with DAPI. A digital slide scanner (3DHISTECH, Hungary) was used to obtain images of the stained section. Image-Pro Plus 6.0 software was used for statistical analysis.

### Statistics and reproducibility

All experiments were repeated independently with similar results at least three times. All experimental data were plotted using Origin 2018 and presented as the mean ± standard deviation (SD). Statistical analyses were performed by SPSS 20.0 using one-way ANOVA with Tukey's multiple comparisons for experiments with three or more groups, differences between two groups were analyzed by the two-tailed unpaired Student's $t$-test. ns: $P > 0.05$, $*P < 0.05$, $**P < 0.01$, and $***P < 0.001$.

### Reporting summary

Further information on research design is available in the Nature Portfolio Reporting Summary linked to this article.

## Data availability

The authors declare that all relevant data supporting the findings of this study are available within this article and its supplementary information. Source data are provided as a Source Data file. Source data are provided with this paper.

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

## Acknowledgements

The authors acknowledge financial support from the National Natural Science Foundation of China T2288101, 11827803, and 12332019 (Y.F.), 32371405 (L.L.), 32171345 (X.L.), 32201114 (Z.M.), the Beijing Natural Science Foundation L248028 (X.L.), the Hebei Provincial Natural Science Foundation of China C2022104003 (X.L.), the Beijing Nova Programme Interdisciplinary Cooperation Project 20230484464 (X.L.), the Shenzhen Science and Technology Program JCYJ20230807095116032 (X.B.), the Fok Ying Tung Education Foundation 141039 (X.L.), the International Joint Research Center of Aerospace Biotechnology and Medical Engineering, Ministry of Science and Technology of China (Y.F.), and the 111 Project B13003 (Y.F.).

## Author contributions

X.B., Z.M., J.G., Y.Z. (YF. Z.), L.L., X.L., and Y.F. designed the research. X.B. and Z.M. fabricated the materials. X.B., Z.M., Y.Z. (YL. Z.), L.Y., and S.H. conducted the in vitro and in vivo experiments. X.B., Z.M., L.L., X.L., and Y.F. treated the experimental data. X.B. and Z.M. wrote and edited the manuscript. X.B., Z.M., L.L., X.L., and Y.F. revised and reviewed the manuscript. X.L. and Y.F. supervised and administrated the research.

## Competing interests

The authors declare no competing interests.
