## [Transparent Peer Review file · Nature Communications]

Janus decellularized membrane with anisotropic cell guidance and anti-adhesion silk-based coatings for spinal dural repair

Corresponding Author: Professor Xiaoming Li

Version 0:

Reviewer comments:

Reviewer #1

(Remarks to the Author)

In this work, the author developed a Janus SIS for the repair of spinal dural defects. Studies including morphology, interfacial adhesion strength, in vitro and in vivo stabilities, anti-adhesion behavior were investigated. However, the evaluation of tissue repair was not convincing.

1. The valid numbers for FT-IR and mechanical properties should be checked. For example, tensile strain should be 197% instead of 196.63%; 1644.37 cm⁻¹ should be 1644 cm⁻¹.
2. The swelling ratio of SiIMA microgroove coating in PBS for different incubation times should be analyzed to demonstrate the stability of the coating.
3. SEM results should be added to reveal the change of the microstructure of SiIMA microgroove coating before and after annealing.
4. For in vitro cell viability study, why the cell proliferation with the treatment of SiIMA microgroove was lower than that by SIS and SiIMA-SIS? It is recommended to use neurons instead of NIH3T3 fibroblasts for in vitro evaluation.
5. For in vivo application, how were the materials fixed on the defects? Did the materials have native adhesion ability with spinal tissues?
6. In Fig 9e, the fluorescence intensity for SMA in Janus SIS group seemed to be higher than that in SIS group.
7. It was hard to see the differences among different groups in Figure 9b, and the author should mark the specific area. Also, the physical properties of the new-formed tissues mediated by SIS and Janus SIS should be compared with the normal dura mater, such as mechanical properties.

Reviewer #2

(Remarks to the Author)

The authors present a study showing that the light responsive silk-based hydrogel coatings fabricated on small intestinal submucosa can effectively promote orientated cell growth by anisotropic patterns, while prohibit fibrotic tissue adhesion on the other side for spinal dural defect repair. Overall, this work is interesting and important to the related fields. The methodology is sound and the work generally meets the expected standards in field of biomedical materials and tissue engineering. However, there are some concerns the authors should address before publication.

1. what is the advantages of anisotropic patterns for spinal dural defect repair?
2. What is the molecular weight of the photocured silk protein?
3. What is the density of UV light used for photo-crosslinking the SiIMA and HAMA?
4. Why does the addition of SiIMA can significantly improve the adhesion strength?

5. The author described that the interior of the SIS membrane was carved with microgrooves at 20 μm intervals and depth for promoting host tissue/cell directional growth. Does the orientated cell growth can only be achieved on microgrooves at 20 μm ? What about other sizes of microgrooves, e.g. 50 μm , 100 μm or even smaller size?
6. Videos on implanting Janus SIS membrane in vivo for repairing spinal dural defects were requested.
7. It is not clear how the spinal tissues have been sectioned, which is the slice thickness, and how many sections and at what distance between them have been used for each immunohistochemistry and histological analysis.
8. Was spinal dura mater defect rat model established by following the procedure from previous literatures? If yes, references should be included properly.

Version 1:

Reviewer comments:

Reviewer #1

(Remarks to the Author)

Thanks for the consideration of all the comments and suggestions. I think the current version is suitable for publication.

Reviewer #2

(Remarks to the Author)

The manuscript can be accepted for publication in its current form.

Response to Reviewer #1:

In this work, the author developed a Janus SIS for the repair of spinal dural defects. Studies including morphology, interfacial adhesion strength, in vitro and in vivo stabilities, anti-adhesion behavior were investigated. However, the evaluation of tissue repair was not convincing.

Response: Thanks very much for your nice comment on this work. We have tried our best to address and improve the evaluation of tissue repair of this work. As suggested, we have repeated the animal experiments and the evaluation of tissue repair has been thoroughly optimized and analyzed in the revised manuscript with red font, including the MRI image (Supplementary Fig. 9), histological analysis (Fig. 9) of overall and specific defect area.

Supplementary Fig. 9. The MRI images after 8 weeks after surgery. MRI images of the spine section 8 weeks after surgery with **a** sagittal and **b** transverse section. (Red arrow refers to the adhesion of the scar tissue to the spinal cord and the infiltration in the spinal canal.)

Fig. 9 | In situ dural mater regeneration enhanced by Janus SIS in a rat spinal dural defect model. (a) Experiment design include MRI, histological analysis and immunology evaluation. (b) macroscopic photographs of spinal tissues 8 weeks after operation, including the normal control (without injury), control group, SIS group and Janus SIS group. (c, d), Histological evaluation (H&E and Masson's trichrome staining) of spinal dural mater repair in different groups. The blue arrow indicates the normal spinal dura mater. The red arrow indicates newborn dura mater-like tissue. The

green arrow indicates fibrotic tissue. The black arrow indicates residual SilMA microgrooves coating. (e) Immunohistochemical staining of α -SMA and collagen I at the focal area of epidural tissues in different groups. (f) Analysis of newborn collagen tissue in the defect area. (g, h) Analysis of α -SMA positive cells and collagen I expression in histological sections. One-way analysis of variance (ANOVA) with a Tukey's post hoc test for multiple comparisons. Values in **f**, **g** and **h** represent the mean \pm SD (n = 3 independent experiments). Source data and exact *P*-values are provided as a source data file. (**P* < 0.05, ***P* < 0.01, ****P* < 0.001).

“H&E and Masson's trichrome staining showed that SIS group presented irregular and discontinuous newly formed collagen tissue was observed in defect areas. Continuous newborn collagen tissue was observed in the dura defect area implanted with Janus SIS group, showing high similarity to normal dura tissue. Moreover, the collagen tissue integrated well with the normal dura tissue. On the contrary, there was almost none newborn collagen tissue in the control group without materials implantation due the lack of regeneration ability of spinal dura mater (Fig. 9c, d, f). Magnetic resonance imaging (MRI) was used to observe the degree of scar adhesion and fibrosis of epidural tissue 8 weeks after operation. As shown in Supplementary Fig. 9, in the control group, severe scar fibrosis in the surgical area could be observed, and the local scar fibrous tissue was found to immerse in the spinal cord. In the SIS and Janus SIS group, the MRI image exhibited a relatively low signal between the spinal cord and the surrounding tissue, indicating no obvious infiltration of epidural scar tissue in the spinal canal. Compared with SIS group, the Janus SIS showed a clearer boundary between spinal cord and the epidural tissue. We further performed immunofluorescence staining of α -SMA and collagen I to assess epidural fibrosis adhesion. The results showed that a large number of α -SMA-positive cells were clustered at the epidural tissue in the control group, and the number of α -SMA-positive fibroblasts was significantly lower in the SIS membrane and Janus SIS groups. Additionally, the Janus SIS group showed fewer α -SMA-positive cells within the epidural tissue compared to the SIS group, it could be inferred that the polyanionic ligand effect of HA inhibited the proliferation and migration of fibroblasts and the expression of postoperative fibrosis related cytokines^{11,27}. In addition, extensively dense collagen I occupied the epidural area in the control group compared with the SIS and Janus SIS groups, while significantly lower collagen I expression could be found in groups implanted with SIS and Janus SIS group. (Fig. 9e, g, h).”

1. The valid numbers for FT-IR and mechanical properties should be checked. For example, tensile strain should be 197% instead of 196.63%; 1644.37 cm^{-1} should be 1644 cm^{-1} .

Response: Thanks for your valuable comment. As suggested, the valid numbers for FT-IR and mechanical properties have been thoroughly checked and optimized in the manuscript with red font. “In the spectra of SilMA microgroove coatings without and with 2 min water vapor annealing treatment, the peaks at 1644 cm^{-1} and 1639 cm^{-1} in the amide I band corresponded to random coils. However, in the spectra after 30 min water vapor annealing, the peak was observed at 1620 cm^{-1} , which is the main absorbance of the β -sheet in amide I.”

“Young’s modulus and toughness than the SilMA coating without water annealing (Fig. 3g, h), with a tensile strength of 12.69 ± 2.06 KPa (increased from 4.39 ± 0.56 KPa), break strain of 197% (increased from 132%), Young’s modulus of 10.58 ± 0.68 KPa (increased from 5.1 ± 1.98 KPa), and toughness of 14.74 ± 0.78 KJ/m² (increased from 3.32 ± 0.2 KJ/m²).”

2. The swelling ratio of SilMA microgroove coating in PBS for different incubation times should be analyzed to demonstrate the stability of the coating.

Response: Thanks for your valuable comment. As suggested, we attempted to measure the swelling rate of the SilMA microgroove coating, but due to the extremely thin coating of the SilMA microgroove on the SIS surface, approximately 20 μm thick. It is difficult to accurately determine the swelling ratio via weight changes between treated and untreated microgrooves due to the SIS substrate is a loose and porous material that is affected by water absorption. Previous extensive studies have illustrated that silk material structure could be regulated by temperature-controlled water vapor annealing. With the increase of water annealing treatment time, the content of β -sheet increased. β -sheets crystals are responsible for the crosslinks, contributing to higher mechanical and structural integrity [1-3].

[1] Sahoo, J.K., Hasturk, O., Falcucci, T. & Kaplan, D.L. Silk chemistry and biomedical material designs. *Nat. Rev. Chem.* **7**, 302-318 (2023).

[2] Reizabal, A., Costa, C. M., Pérez-Alvarez, L., Vilas-Vilela, J. L. & Lanceros-Méndez, S. Silk Fibroin as Sustainable Advanced Material: Material Properties and Characteristics, Processing, and Applications. *Adv. Funct. Mater.* DOI10.1002/adfm.202210764 (2023).

[3] Hu, X., Shmelev, K., Sun, L., Gil, E.S., Park, S.H., Cebe, P. & Kaplan, D. L. Regulation of Silk Material Structure by Temperature-Controlled Water Vapor Annealing. *Biomacromolecules*.**12**, 1686-1696 (2011).

3. SEM results should be added to reveal the change of the microstructure of SilMA microgroove coating before and after annealing.

Response: Thanks for your valuable comment. As suggested, the microstructure of SilMA microgroove coating before and after annealing have been added accordingly in the manuscript with red font (Fig. 2). The results showed that the microstructure of SilMA microgroove coating did not change before and after annealing.

“In the SilMA microgroove coating samples, the interior of the SilMA-SIS membrane was carved with precise microgrooves at 20 μm intervals and depths, and there were no changes on the microstructure of SilMA microgroove coating after water annealing.”

4. For in vitro cell viability study, why the cell proliferation with the treatment of SilMA

microgroove was lower than that by SIS and SilMA-SIS? It is recommended to use neurons instead of NIH3T3 fibroblasts for in vitro evaluation.

Response: Thanks for your valuable comment and advice. The main reason for the lower cell proliferation with the treatment of SilMA microgroove is that, cell morphology on the anisotropic microgroove showing significantly elongated morphology and a smaller spreading area relative to SilMA microgroove. Previous studies illustrated that cell morphology changes were believed to be highly associated with cell proliferation. For example, both aligned micro/nanogrooved and electrospun fiber substrate and carbon nanotubes could modulate cell shape and suppress fibroblast proliferation [4-5]. These results revealed the contact guidance effect from anisotropic topology on cellular morphology and further on the proliferation activity of cells.

The main reason for the evaluation of NIH3T3 fibroblasts in vitro was that the spinal dura mater mainly contains collagenous fibers and fibroblasts [6]. In the context of dura mater damage, fibroblasts play intricate roles in both dura regeneration and fibrosis formation. The goal of repairing damaged dura tissue not only lies in promoting natural healing of dura mater through promoting fibroblast activity but also asks for effective prevention of fibrosis, which is derived from excessive ECM component (collagen) deposition secreted by activated and proliferated fibroblasts [4]. After comprehensive consideration, it may be more appropriate to select NIH3T3 fibroblasts for in vitro evaluation in this study.

[4] Xu, Y., Shi, G.D., Tang, J.C., Cheng, R.Y., Shen, X.F., Gu, Y., Wu, L., Xi, K., Zhao, Y.H., Cui, W.G. & Chen, L. ECM-inspired micro/nanofibers for modulating cell function and tissue generation. *Sci. Adv.* **6**, eabc2036 (2020).

[5] Weng, W.Z., He, S.S., Song, H.Y., Li, X.Q., Cao, L.H., Hu, Y.J., Cui, J., Zhou, Q.R., Peng, H.S. & Su, J.C. Aligned Carbon Nanotubes Reduce Hypertrophic Scar via Regulating Cell Behavior. *ACS nano*, **12** (2018).

[6] Protasoni, M., Sangiorgi, S., Cividini, A., Culivaris, G.T., Tomei, G., Dell'Orbo, C., Raspanti, M., Balbi, S. & Reguzzoni, M. The collagenic architecture of human dura mater. *J. neurosurg.*, **114**, 1723-1730 (2011).

5. For in vivo application, how were the materials fixed on the defects? Did the materials have native adhesion ability with spinal tissues?

Response: Thanks for your valuable comment. The material is naturally physically adhered to the defect site (Supplementary Figure 8e). The main reason for this operation is that the material itself is not a dense structure, and it is also a relatively soft film material after absorbing water. Therefore, when the dry material is implanted, it becomes soft and easily conforms to tissue when it is soaked in blood. In addition, the size of implant material is larger than the defect area, the material periphery can be inserted into the muscle gap to further strengthen the fixation. The implantation process can be seen in the Supplementary video 1 provided. The detail of surgical procedure is demonstrated as follows Supplementary Figure 8. On the other hand, both SIS and Janus SIS, the surface of the material is rich in amino acids, which have a certain affinity with the amino/carboxyl groups on the surface of tissues. Overall, we believe that the primary means of fixation is achieved by the structural characteristics of the material itself for physical adhesion, while also utilizing the restriction of surrounding muscles to secure the material in place at the defect site.

Supplementary Figure 8. The detail of surgical procedure. a. A midline skin incision was made; b. The spinous process was exposed; c. The spinous process was cut; d. The laminectomy was performed and the excision of dura mater was made. **e. Dural defect covered with a Janus SIS membrane.** f. The muscle and skin were sutured. The black dashed lines indicated the dural defect area and the implanted material.

6. In Fig 9e, the fluorescence intensity for SMA in Janus SIS group seemed to be higher than that in SIS group.

Response: Thanks for your valuable comment. The animal experiments were performed again and the evaluation of tissue repair has been thoroughly optimized in the revised manuscript with red font. The results showed that the fluorescence intensity for α -SMA in Janus SIS group is lower than SIS group. Additionally, the MRI image could more directly observe the degree of fibrosis adhesion in Janus SIS group is lower than that in SIS group.

Fig. 9e. Immunohistochemical staining of α -SMA at the focal area of epidural tissues in different groups after 8 weeks of implantation.

Supplementary Fig. 9. The MRI images after 8 weeks after surgery. MRI images of the spine section 8 weeks after surgery with **a** sagittal and **b** transverse section. (Red arrow refers to the adhesion of the scar tissue to the spinal cord and the infiltration in the spinal canal.)

7. It was hard to see the differences among different groups in Figure 9b, and the author should mark the specific area. Also, the physical properties of the new-formed tissues mediated by SIS and Janus SIS should be compared with the normal dura mater, such as mechanical properties.

Response: Thanks for your valuable comment and advice. As suggested, Figure 9b has been optimized, with special areas marked and clearly explained. Furthermore, the evaluation of tissue repair has been thoroughly optimized in the revised manuscript in the manuscript with red font (Supplementary Fig. 9 and Fig. 9), including the MRI image, histological and immunofluorescence analysis of overall and specific defect area. As for the physical properties of the new-formed tissues between the different groups, we have tried our best to measure the mechanical properties. However, we apologize that the dura mater of the rat is too fragile to complete the mechanical test. In future studies, we will attempt to use large animal models to test the mechanical properties of regenerated and normal dura mater.

Fig. 9b. Photographs of spinal tissues after 8 weeks of repair in the rat spinal dural defect model, including the natural dura mater, injury group (control), SIS repair group and Janus SIS repair group.

Response to Reviewer #2:

The authors present a study showing that the light responsive silk-based hydrogel coatings fabricated on small intestinal submucosa can effectively promote orientated cell growth by anisotropic patterns, while prohibit fibrotic tissue adhesion on the other side for spinal dural defect

repair. Overall, this work is interesting and important to the related fields. The methodology is sound and the work generally meets the expected standards in field of biomedical materials and tissue engineering. However, there are some concerns the authors should address before publication.

Response: Thank you very much for your affirmation and nice comment on this work. We have tried our best to address these concerns and improve the clarity of this work.

1. what is the advantages of anisotropic patterns for spinal dural defect repair?

Response: Thanks for your valuable comment. The natural spinal dura mater extracellular matrix (ECM) contains collagenous fibers, which are arranged in parallel along the longitude direction. Inspired by the peculiar nature of spinal dura matter, we developed anisotropic patterns decorated Janus SIS with biomimetic features to regulate cell behavior for better reproduction of the heterogeneous ECM microstructure for dura mater repair.

The anisotropic scaffolds have been used successfully in various tissue engineering applications, such as hierarchical anisotropic (skeletal muscle, peripheral nerve, tendon, and ligament), plate-like anisotropic (myocardium, corneal stroma, meniscus, and articular cartilage), and tubular-shaped anisotropic (annulus fibrosus and smooth muscle).[1]

[REDACTED]

Fig.1 (A) 3D-Printed Biomimetic Scaffold efficiently promote regeneration of muscle cells; (B) Formation of composite cryogels with anisotropic and homogeneous porous structures, which mimic the hierarchical structures of cortical and cancellous bones. (The figures taken from the literature [2-3])

[1] Xing, J.Y., Liu, N., Xu, N.N., Chen, W.J. & Xing, D.M. Engineering Complex Anisotropic Scaffolds beyond Simply Uniaxial Alignment for Tissue Engineering. *Adv. Funct. Mater.* **32**, 2110676 (2022).

[2] Kim, W.J., Kim, M. & Kim, G.H. 3D - Printed Biomimetic Scaffold Simulating Microfibril Muscle Structure. *Adv. Funct. Mater.* DOI10.1002/adfm.201800405 (2018).

[3] Fan, Z.H., Liu, H.X., Ding, Z.Z., Xiao, L.Y., Lu, Q. & Kaplan, D.L. Simulation of Cortical and Cancellous Bone to Accelerate Tissue Regeneration. *Adv. Funct. Mater.* DOI10.1002/adfm.202301839 (2023).

2. What is the molecular weight of the photocured silk protein?

Response: Thanks for your valuable comment. The weight-average molecular weight of the photocured silk protein is about 10.632KDa, the detailed content has been supplied to the

Supplementary Materials figure 2.

3. What is the density of UV light used for photo-crosslinking the SilMA and HAMA?

Response: Thanks for your valuable comment. The density of UV light used for photo-crosslinking the SilMA and HAMA was $25\text{mW}/\text{cm}^2$. The relevant content has been supplemented and illustrated in the experimental methods section with red font.

“The SilMA-SIS was covered on the SilMA solution followed by photocuring with UV irradiation (405 nm) for 35 s at the density of $25\text{mW}/\text{cm}^2$ to generate the corresponding microgrooves. Next, on the top surface of the SilMA-SIS, SilMA (0.5% w/v, LAP 0.25%) and HAMA (1% w/v, LAP 0.25%) mixture was smeared uniformly and photo-polymerized under UV light (405 nm, $25\text{mW}/\text{cm}^2$) for 15 s and then drying at $37\text{ }^\circ\text{C}$.”

4. Why does the addition of SilMA can significantly improve the adhesion strength?

Response: Thanks for your valuable comment. According to previous report, the adhesion strength related to interfacial bonding behavior and mechanical strength of materials [4-6]. In this study, methacryloyl-functionalized SF (SilMA) was firstly deposited on the SIS surface via LbL self-assembly, thereafter, the SIS surface is rich in photocrosslinked carbon-carbon double bond groups. The double bonds on the surface of SIS can crosslinked with the SilMA microgrooves functional coating, resulting in covalent bond and increasing the interface bonding strength. As in our previous work, we introduced azido groups into SF molecules during the LbL process to serve as covalent anchors for other functional coatings [7]. As for SilMA-HAMA coating, Figure 5b revealed that the addition of SilMA enhances the anti-swelling ability of HAMA in vivo, in other words, the structural and mechanical stability of HAMA is increased, so the interface adhesion strength will be improved. The detailed content has been added in the manuscript of discussion with red font.

[4] Yang, Y., Xu, T., Bei, H.P., Zhao, Y. & Zhao, X. Sculpting Bio-Inspired Surface Textures: An Adhesive Janus Periosteum. *Adv. Funct. Mater.* **31**, 2104636 (2021)

[5] Walia, R., Akhavan, B., Kosobrodova, E., Kondyurin, A. & Bilek, M.M. Hydrogel-Solid Hybrid Materials for Biomedical Applications Enabled by Surface-Embedded Radicals. *Adv. Funct. Mater.* **30**, 2004599 (2020).

[6] Wu, J. et al. Anti-Swelling, Robust, and Adhesive Extracellular Matrix - Mimicking Hydrogel Used as Intraoral Dressing. *Adv. Mater.* **34**, 2200115 (2022).

[7] Yang, L., Lin, X., Zhou, J., Hou, S. & Fan, Y. Cell Membrane-Biomimetic Coating via Click-Mediated Liposome Fusion for Mitigating the Foreign-Body Reaction. *Biomaterials* 120768 (2021).

5. The author described that the interior of the SIS membrane was carved with microgrooves at 20 μm intervals and depth for promoting host tissue/cell directional growth. Does the orientated cell growth can only be achieved on microgrooves at 20 μm ? What about other sizes of microgrooves, e.g. 50 μm , 100 μm or even smaller size?

Response: Thanks for your valuable comment. It is regrettable that we did not thoroughly compare the guiding behavior of cells on different microgrooves sizes in this study. Previous studies revealed that microgrooves with widths of 15 – 60 μm have contact guidance on the cells [8]. In addition, study has been carried out that the cells seeded on the 30 μm microgrooves exhibited lower inflammatory responses and higher cell adhesion and migration than the 60 μm microgrooves group due to the better contact guidance [9]. In another study, the 20 μm size microgrooves was chosen because it can effectively induce oriented morphology via contact guidance [10]. Additionally, early classic research demonstrated that 20 μm size microgrooves are appropriately sized relative to cells, could effectively promote macrophages to express more Arg1 proteins compared to 50 μm size microgrooves, indicating an effectively transition towards the M2 phenotype [11]. In this study, we attempt to utilize the microgrooves to not only induce the alignment of tissue and cells but also to regulate macrophage polarization to play a positive role in tissue repair and adhesion processes within the injury and regeneration microenvironment. Therefore, based on previous extensive studies, this work creatively constructed a 20 μm intervals and depth microgroove on SIS membrane. In the future, we will comprehensively and deeply compare the guiding behavior of cells and the effects of different groove sizes on spinal dural defect repair.

[8] Lai, Y.Z., Chen, J., Zhang, T., Gu, D.D., Zhang, C.Q., Li, Z.F., Lin, S. & Fu, X.M. Schultze-Mosgau, S., Effect of 3D microgroove surface topography on plasma and cellular fibronectin of human gingival fibroblasts. *J. Dent.* **41**, 1109-1121 (2013).

[9] Sun, J., Ding, Q., Chen, Y., Li, J.J., Wang, Z.H., Wei, Z.Y., Ge, X.Y. & Zhang, L. Effects and underlying mechanism of micro-nano-structured zirconia surfaces on biological behaviors of human gingival fibroblasts under inflammatory conditions. *Acta biomater.* **183**, 356-370 (2024)

[10] Yi, B.C., Zhou, B.Y., Song, Z.F., Yu, L., Wang, W.B. & Liu, W. Step-wise CAG@PLys@PDA-Cu²⁺ modification on micropatterned nanofibers for programmed endothelial healing. *Bioact. Mater.* **25**, 657-676 (2023).

[11] McWhorter, F.Y., Wang, T.T., Nguyen, P., chung, T. & Liu, W.F. Modulation of macrophage phenotype by cell shape. *Proc. Natl. Acad. Sci. U. S. A.* **110**, 17253-17258 (2013).

6. Videos on implanting Janus SIS membrane in vivo for repairing spinal dural defects were requested.

Response: Thanks for your valuable comment. As suggested, the video on implanting Janus SIS membrane in vivo for repairing spinal dural defects have been added in Supplementary Video 1, and the detail of surgical procedure was demonstrated in Supplementary Figure 8.

Supplementary Figure 8. The detail of surgical procedure. a. A midline skin incision was made; b. The spinous process was exposed; c. The spinous process was cut; d. The laminectomy was performed and the excision of dura mater was made. e. Dural defect covered with a Janus SIS membrane. f. The muscle and skin were sutured. The black dashed lines indicated the dural defect area and the implanted material.

7. It is not clear how the spinal tissues have been sectioned, which is the slice thickness, and how many sections and at what distance between them have been used for each immunohistochemistry and histological analysis.

Response: Thanks for your valuable comment. The specimen preparation for histological analysis has been supplemented and illustrated in the Experimental Methods section with red font. The Experimental Methods section have been thoroughly checked and optimized in the manuscript with red font.

“For histological analysis, cross-sections of the spine samples with 5- μ m-thick are made for Hematoxylin and eosin (H&E) staining, Masson's trichrome staining and immunological evaluations. H&E and Masson's trichrome staining were performed to assess the stability of the coating, tissue integration, anti-adhesion performance, collagenous fibrotic capsule formation, and regeneration of the defect dura mater. For immunofluorescence staining, all sections were incubated with primary antibodies against CD68 (pan-macrophage cell surface marker) (1:200, Abcam, USA), CCR7 (M1 marker) (1:200, Abcam, USA), Arg-1 (M2 marker) (1:200, Abcam, USA), α -SMA (1:100, Abcam, USA) and Collagen I (1:50, BOSTER, China) at 4 °C overnight, followed by treatment with Alexa Fluor 488/594-labeled secondary antibody (Solarbio, China) for 1 h at 37 °C. Finally, the samples were rinsed with PBS and stained with DAPI (Solarbio, China) for 15 minutes at room temperature. A digital slide scanner (3DHISTECH, Hungary) was used to obtain images of the stained section. Image-Pro Plus 6.0 software was used for statistical analysis.”

8. Was spinal dura mater defect rat model established by following the procedure from previous literatures? If yes, references should be included properly.

Response: Thanks for your valuable comment. The references have been added accordingly in the manuscript [12].

“For spinal dura mater defect repair of Janus SIS, a rat duraplasty model was prepared according to previous established procedure.”

[12] Xu, Y., Shi, G.D., Tang, J.C., Cheng, R.Y., Shen, X.F., Gu, Y., Wu, L., Xi, K., Zhao, Y.H., Cui, W.G. & Chen, L. ECM-inspired micro/nanofibers for modulating cell function and tissue generation. *Sci. Adv.* **6**, eabc2036 (2020).